# Hard Labels In! Rethinking the Role of Hard Labels in Mitigating Local Semantic Drift

## Abstract

Soft labels generated by teacher models have become a dominant paradigm for knowledge transfer and recent large-scale dataset distillation such as SRe$^2$L, RDED, LPLD, offering richer supervision than conventional hard labels. However, we observe that when only a limited number of crops per image are used, soft labels are prone to local semantic drift: a crop may visually resemble another class, causing its soft embedding to deviate from the ground-truth semantics of the original image. This mismatch between local visual content and global semantic meaning introduces systematic errors and distribution misalignment between training and testing. In this work, we revisit the overlooked role of hard labels and show that, when appropriately integrated, they provide a powerful content-agnostic anchor to calibrate semantic drift. We theoretically characterize the emergence of drift under few soft-label supervision and demonstrate that hybridizing soft and hard labels restores alignment between visual content and semantic supervision. Building on this insight, we propose a new training paradigm, **H**ard Label for **A**lleviating **L**ocal Semantic **D**rift (**HALD**), which leverages hard labels as intermediate corrective signals while retaining the fine-grained advantages of soft labels. Extensive experiments on dataset distillation and large-scale conventional classification benchmarks validate our approach, showing consistent improvements in generalization. On ImageNet-1K, we achieve 41.8% with only 285M storage for soft labels, outperforming prior state-of-the-art LPLD by **8.1%**. Our findings re-establish the importance of hard labels as a complementary tool, and call for a rethinking of their role in *soft-label–dominated* training.

## 1 Introduction

Soft labels have emerged as a standard and strong supervision signal derived from pretrained teacher models in knowledge distillation (Hinton et al., 2015) and dataset distillation (Yin et al., 2023) tasks. Unlike hard labels, which provide only class-level supervision, soft labels encode richer inter-class similarity information, offering smoother gradients and better generalization. In particular, for dataset distillation, FKD-based (Shen & Xing, 2022) soft labels have become indispensable because they allow student models to inherit semantic knowledge from powerful teacher networks without relying on access to the teacher in the post stage. This is especially critical in the post-training stage, where the teacher must be completely isolated from the training pipeline as it is trained on the original data, to avoid information leakage and any direct contact with raw full data following the task setting.

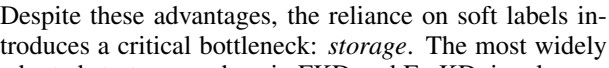

Figure 1: Illustration of local-view semantic drift: partial crops may change object–label relations, yielding semantics that deviate from the full image.

Despite these advantages, the reliance on soft labels introduces a critical bottleneck: *storage*. The most widely adopted strategy, such as in FKD and FerKD, involves pre-computing and saving soft labels for every image crop in the distilled dataset. While effective, this design leads to massive storage requirements, particularly on large-scale datasets like ImageNet-1K (*distilled data*: 750M vs. *soft-label*:

28.33 GB), even larger than the distilled data storage size which is not acceptable. Thus, storing per-crop logits across thousands of classes results in prohibitive memory overhead, hindering the scalability and practical deployment of dataset distillation pipelines. As datasets continue to grow in size and granularity, compressing or reducing soft label storage has become an urgent problem.

A straightforward approach to alleviate storage costs is to reduce the number of crops, and consequently, the number of soft labels per image. However, this seemingly simple solution introduces a more subtle and overlooked issue: *local semantic drift*. As shown in Fig. 1, since crops often capture only partial or ambiguous regions of an image, their soft labels may semantically shift toward unrelated categories. For example, a crop from a cat image might be similar to a rabbit, and the soft embedding derived from the teacher would misalign with the global semantics of the original class. This mismatch between local visual evidence and global semantics undermines training, leading to degraded generalization and unstable predictions. We also provide a theoretical guarantee by establishing a strictly positive lower bound on the expected mismatch between the objective defined with reduced crops and that with sufficient crops. Our analysis shows that this gap is inversely proportional to the number of crops: *the fewer the crops, the larger the mismatch*.

**Hard labels as a corrective signal.** In contrast, hard labels are content-agnostic and immune to local visual ambiguity. While they lack the fine-grained information encoded in soft labels, they provide a stable supervisory anchor tied to the ground-truth semantic identity of the image. This raises a key insight: hard labels, if carefully integrated, could serve as corrective signals to calibrate soft-label supervision and mitigate semantic drift. Surprisingly, this potential has been largely overlooked in the literature, where hard labels are often considered too coarse or discarded entirely in favor of soft labels. From a theoretical perspective, we further guarantee that proper joint training with soft and hard labels does not introduce gradient inconsistencies that would hinder optimization. On the contrary, the controlled fluctuations arising from hard-label supervision inject additional information, boosting the learning of new knowledge beyond what soft labels alone can provide.

**Our contributions.** In this work, we revisit the role of hard labels in dataset distillation and propose a hybrid training paradigm, **H**ard Label for **A**lleviating **L**ocal Semantic **D**rift (**HALD**). The core idea is to strategically integrate hard labels into the training pipeline, using them to calibrate and realign the semantic space of crops while preserving the nuanced information provided by soft labels. We provide theoretical analysis showing why few soft-label inevitably causes semantic drift, and demonstrate mathematically how hard labels can compensate for this effect. Building on this foundation, we validate **HALD** through extensive experiments across multiple benchmarks, consistently showing that it reduces distribution mismatch and improves generalization, even under *aggressive soft-label compression*.

## 2 RELATED WORK

**Dataset Distillation.** Dataset distillation aims to construct a small, synthetic surrogate of a large dataset that retains its core information content. The goal is to accelerate training and cut storage costs while achieving performance close to training on the full data. Current approaches can be broadly grouped into six families: 1) *Gradient Matching* (Zhao et al., 2020; Zhao & Bilen, 2021; Lee et al., 2022; Kim et al., 2022; Zhou et al., 2024). 2) *Meta-Model Matching* (Wang et al., 2018; Nguyen et al., 2021; Loo et al., 2022; Zhou et al., 2022; Deng & Russakovsky, 2022; He et al., 2024). 3) *Trajectory Matching* (Cui et al., 2023; Chen et al., 2023; Guo et al., 2024). 4) *Distribution Matching* (Wang et al., 2022; Zhao & Bilen, 2023; Xue et al., 2024; Lee et al., 2022; Sajedi et al., 2023; Shin et al., 2024; Liu et al., 2022). 5) *Decoupled Optimization* (Yin et al., 2023; Shao et al., 2024a;b; Yin & Shen, 2024; Zhang et al., 2025; Cui et al., 2025a; Tran et al., 2025; Shen et al., 2025; Sun et al., 2024). 6) *Difussion Based* (Gu et al., 2024; Su et al., 2024; Chen et al., 2025; Zhao et al., 2025; Chan-Santiago et al., 2025; Wang et al., 2025; Zou et al., 2025). A comprehensive overview of recent advances can be found in (Liu & Du, 2025; Shang et al., 2025).

**Soft Label and Hard Label Usage.** Soft labels are widely adopted in dataset distillation for their richer target structure relative to hard labels, enabling finer guidance during optimization (Yin et al., 2023; Qin et al., 2024; Sun et al., 2024; Yu et al., 2024; Cui et al., 2025b). However, storing per-sample soft targets can introduce a substantial memory overhead, often comparable to or larger than the image storage itself. To mitigate this, LPLD (Xiao & He, 2024) proposes generating a limited set of soft targets and reusing them throughout training, substantially reducing the label-storage budget.

Yu et al. (2024) proposes a label-lightening framework HeLlO that leverages effective image-to-label projectors to directly generate synthetic labels online from synthetic images. In parallel, several works revisit the role of hard labels in distillation: EDC (Shao et al., 2024b) combines hard- and soft-label objectives within a unified loss to improve performance, while GIFT (Shang et al., 2024) fuses hard information into soft targets to obtain more reliable supervision.

## 3 APPROACH

### 3.1 PRELIMINARY

**Dataset Distillation/Condensation.** Given dataset $\mathcal{O} = \{(x_i, y_i)\}$, dataset distillation seeks a small set $\mathcal{C} = \{(\tilde{x}_j, \tilde{y}_j)\}$ ($|\mathcal{C}| \ll |\mathcal{O}|$) such that models trained on $\mathcal{C}$ and $\mathcal{O}$ generalize similarly:

$$\min_{\mathcal{C}} \sup_{(x,y)\sim\mathcal{O}} |\mathcal{L}(f_{\theta_{\mathcal{O}}}(x), y) - \mathcal{L}(f_{\theta_{\mathcal{C}}}(x), y)|. \tag{1}$$

Here, $\theta_{\mathcal{O}}$ and $\theta_{\mathcal{C}}$ are obtained via ERM (Vapnik, 1991) on $\mathcal{O}$ and $\mathcal{C}$, respectively. With this setup in place, prevailing evaluations of distilled datasets rely on pre-generated soft labels, which tends to underemphasize the role of hard labels, despite their zero storage cost and direct ground-truth supervision. Moreover, storing soft labels (often per crop/augmentation) can exceed the images themselves, motivating storage-efficient alternatives. We therefore revisit this design choice and analyze the consequences of a soft-only protocol, particularly under limited soft-label coverage.

**Soft Label Recap.** Using a teacher model to generate soft labels (Hinton et al., 2015) for training a new model has become both common and popular, especially in the field of dataset distillation (Wang et al., 2018), where it has repeatedly been shown to be particularly effective and important for large-scale datasets. The main drawback of soft labels, however, is that each crop requires storing its own soft label, which leads to substantial storage overhead (Yin et al., 2023). A straightforward workaround is to reduce the number of crops (and thus soft labels) per image (Xiao & He, 2024).

However, we identify an often-overlooked issue that arises when only a small number of soft labels are used per image: *Semantic Shift*. As illustrated in Fig. 1, soft labels are usually assigned to image crops, but these crops may only capture partial regions. This semantic shift problem is intrinsic to soft labels, whereas hard labels, being content-agnostic, do not suffer from such drift. While hard labels bring their own limitation: they fail to align the semantic label with the fine-grained visual content, making it difficult for the model to learn detailed representations.

Our work addresses precisely this trade-off. In the following sections, we first provide a theoretical analysis showing why using too few soft labels per image introduces a semantic shift, leading to mismatched train–test distributions and degraded predictions. We then demonstrate how, when used appropriately, hard labels can serve as a corrective signal to calibrate this mismatch, since they provide supervision independent of crop content. Finally, we propose a new training paradigm, *Soft–Hard–Soft*, and show through both theoretical explanation and empirical visualizations that it effectively resolves the limitations of existing approaches.

### 3.2 TRAINING WITH LIMITED SOFT LABEL COVERAGE

**Definition 1** (Local-View Semantic Drift (LVSD)). *Fix $\tilde{x}$ and an augmentation distribution $\mathcal{T}(\tilde{x})$. For a random crop $x^{(\text{crop})} \sim \mathcal{T}(\tilde{x})$, let $\tilde{p}(x^{(\text{crop})}) \in \Delta^C$ be the teacher's soft prediction and write*

$$\bar{p} := \mathbb{E}[\tilde{p}(x^{(\text{crop})})], \qquad \Sigma := \text{Cov}[\tilde{p}(x^{(\text{crop})})].$$

*We say the supervision exhibits* Local-View Semantic Drift (LVSD) *iff $\Sigma \neq 0$.*

**Lemma 1.** *For $s$ i.i.d. crops define $\hat{p}_s := \frac{1}{s}\sum_{i=1}^{s} \tilde{p}(x_i^{(\text{crop})})$. Then,*

$$\mathbb{E}[\hat{p}_s] = \bar{p}, \qquad \text{Cov}(\hat{p}_s) = \frac{\Sigma}{s}, \qquad \mathbb{E}\big[\|\hat{p}_s - \bar{p}\|_2^2\big] = \frac{\text{Tr}(\Sigma)}{s}.$$

*In particular, under LVSD, the deviation is strictly positive for any finite $s$ and decays as $\mathcal{O}(1/s)$.*

**Definition 2** (Soft Label per Image (SLI)). *The soft labels per image (SLI) denote the number of augmented soft labels (e.g., crops or views) generated for each image.*

**Definition 3** (Soft Label per Class (SLC)). *Let $C \in \mathbb{N}$ be the number of classes, and* ipc *the number of images per class. Given that each image has* SLI *soft labels, the total number of soft labels per class (SLC) is defined as*

$$\text{SLC} = \text{ipc} \times \text{SLI}.$$

*Each soft label is a $C$-dimensional vector, with each scalar entry stored using $b$ bits. The corresponding per-class storage budget (in bits) is therefore*

$$\text{Storage(SLC)} = \text{SLC} \cdot (Cb).$$

***Discussion (Why SLC).*** SLC quantifies per-class pre-generated supervision. Fixing SLC controls label-side storage: regardless of IPC, equal SLC yields the same number of stored soft labels per class. By contrast, pruning ratios alone are confounded by IPC and obscure absolute storage.

To reduce the storage overhead associated with soft-label supervision, LPLD Xiao & He (2024) proposes limiting the total number of stored teacher predictions and reusing them across training. We refer to this storage budget as SLC (see Definition 3). While this reduces storage substantially, Lemma 1 implies that finite-$s$ supervision deviates from the full-coverage regime due to *Local-View Semantic Drift* (see Definition 1). In what follows, we quantify this deviation.

**Deviation from the Ideal Optimization Objective.** By Theorem 1, *Local-View Semantic Drift (LVSD)*, i.e., nonzero per-crop prediction covariance, induces a *strictly positive* lower bound on the expected mismatch between $\mathcal{L}_s$ and $\mathcal{L}_{\text{ideal}}$ of order $\Theta(s^{-1/2})$, with a distribution-dependent constant $C(\sigma, \kappa)$. Consequently, in low-SLC regimes (small $s$) the finite-SLC objective is systematically misaligned with the ideal supervision goal, the gap decays and vanishes only as $s \to \infty$.

**Theorem 1** (Proof in Appendix B.1). *Consider a synthetic image $\tilde{x}$ with augmentation distribution $\mathcal{T}(\tilde{x})$. Each crop $\tilde{x}_i^{(\text{crop})} \sim \mathcal{T}(\tilde{x})$ is assigned a teacher soft label $\tilde{p}_i \in \Delta^C$, while the student model produces a predictive distribution $q_\theta(\cdot \mid \tilde{x}_i^{(\text{crop})})$. Let $\mathcal{L}[\tilde{p}, q] : \Delta^C \times \Delta^C \to \mathbb{R}_{\geq 0}$ be a per-crop loss functional. The empirical training loss based on $s$ independent crops is $\mathcal{L}_s(\theta; \tilde{x}) = \frac{1}{s}\sum_{i=1}^{s} \mathcal{L}\big[\tilde{p}_i, q_\theta(\cdot \mid \tilde{x}_i^{(\text{crop})})\big]$, while the ideal loss under full augmentation coverage is $\mathcal{L}_{\text{ideal}}(\theta; \tilde{x}) = \mathbb{E}_{\tilde{x}^{(\text{crop})} \sim \mathcal{T}(\tilde{x})}\big[\mathcal{L}\big[\tilde{p}, q_\theta(\cdot \mid \tilde{x}^{(\text{crop})})\big]\big]$. Assume that the per-crop loss has finite variance and finite fourth central moment:*

$$\sigma^2 = \text{Var}_{\tilde{x}^{(\text{crop})} \sim \mathcal{T}(\tilde{x})}\Big[\mathcal{L}\big[\tilde{p}, q_\theta(\cdot \mid \tilde{x}^{(\text{crop})})\big]\Big] < \infty, \quad \kappa = \frac{\mathbb{E}\Big[\big(\mathcal{L} - \mathbb{E}\mathcal{L}\big)^4\Big]}{\sigma^4} \in [1, \infty).$$

*Then the expected deviation between empirical and ideal losses satisfies:*

$$\mathbb{E}\big[\big|\mathcal{L}_s(\theta; \tilde{x}) - \mathcal{L}_{\text{ideal}}(\theta; \tilde{x})\big|\big] \geq \frac{\sigma}{\sqrt{s}} \cdot \frac{16}{25\sqrt{5}} \cdot \min\big\{\tfrac{1}{\kappa}, \tfrac{1}{3}\big\}. \tag{2}$$

**Few Soft Labels Make Train-Test Misaligned.** Let $\hat{\theta}_\star := \arg\min_\theta \mathcal{L}_{\text{ideal}}(\theta)$ denote the *oracle* obtained under exhaustive local-view supervision from a strong teacher. By construction, $\hat{\theta}_\star$ *maximally aligns* with the teacher's predictive distribution across local views, we assume it achieves the best attainable generalization. Thus any deviation $\hat{\theta}_s \neq \hat{\theta}_\star$ may degrade generalization. We therefore study the excess loss $\mathbb{E}\big[\mathcal{L}_{\text{ideal}}(\hat{\theta}_s) - \mathcal{L}_{\text{ideal}}(\hat{\theta}_\star)\big]$, which is non-negative by the optimality of $\hat{\theta}_\star$ for $\mathcal{L}_{\text{ideal}}$ and vanishes iff $\hat{\theta}_s = \hat{\theta}_\star$. Under limited soft-label coverage (small $s$), $\mathcal{L}_s$ exhibits *LVSD* and optimizes a proxy of $\mathcal{L}_{\text{ideal}}$; consequently $\hat{\theta}_s$ departs

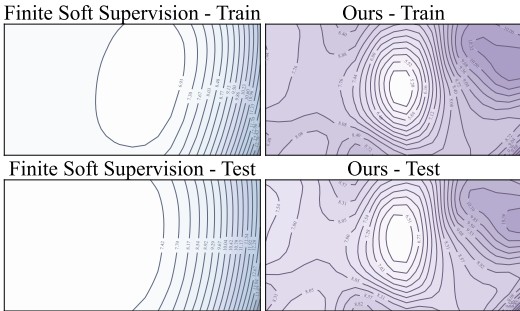

Figure 2: Train and test loss landscapes on an IPC=10 distilled dataset with SLC=50, comparing (i) finite soft-label coverage and (ii) our method.

from $\hat{\theta}_\star$, incurring an unavoidable generalization penalty. Theorem 2 formalizes this effect, yielding a lower bound of order $\Omega(1/s)$ that decays sublinearly and disappears only as $s \to \infty$.

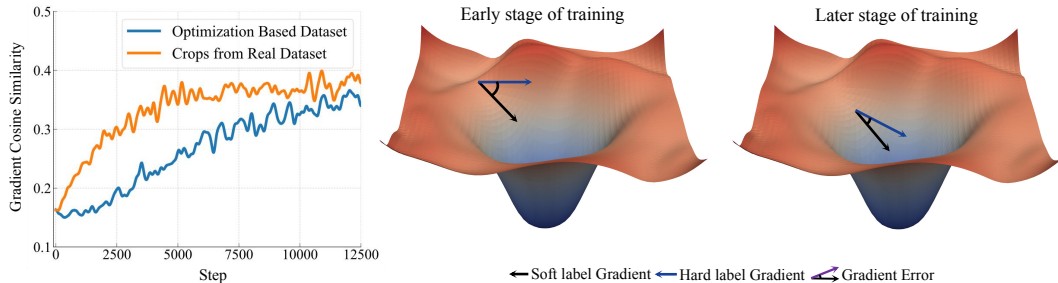

Figure 3: Gradient similarity between hard- and soft-label losses over training, evaluated on real-image crops and optimization-based distilled data, showing a clear upward trend indicative of strengthened alignment.

**Theorem 2** (Proof in Appendix B.2). *Let $\mathcal{L}_{ideal}(\theta)$ be twice continuously differentiable in a neighborhood $\mathcal{N}$ of its unique minimizer $\hat{\theta}_{\star}$, and denote $H_{\star} := \nabla^2 \mathcal{L}_{ideal}(\hat{\theta}_{\star}) \succeq \mu I$ for some $\mu > 0$. Write $g(\theta; x) := \nabla_{\theta} \ell(\theta; x)$ so that $\nabla \mathcal{L}_{ideal}(\theta) = \mathbb{E}[g(\theta; x)]$, and let $\hat{\theta}_s \in \arg\min_{\theta} \mathcal{L}_s(\theta)$ be any ERM. Assume: (A1) (Unbiased score & covariance) $\mathbb{E}[g(\hat{\theta}_{\star}; x)] = 0$, and $\Sigma_{\star} := \mathrm{Cov}(g(\hat{\theta}_{\star}; x))$ with $\mathbb{E}\|g(\hat{\theta}_{\star}; x)\|^{2+\kappa} < \infty$ for some $\kappa > 0$. (A2) (Hessian Lipschitz) $\nabla^2 \mathcal{L}_{ideal}$ is $L_H$-Lipschitz on $\mathcal{N}$. (A3) (Local uniform concentration) There exist $r_0 > 0$ and constants $C_{uc} > 0$, $\bar{c} > 0$ such that for all $s$ and $\delta \in (0, 1)$, with probability at least $1 - \delta$, $\sup_{\theta \in \mathbb{B}(\hat{\theta}_{\star}, r_0)} \left\| H_s(\theta) - \nabla^2 \mathcal{L}_{ideal}(\theta) \right\| \leq C_{uc}\sqrt{\frac{\log(1/\delta)}{s}}$, $H_s(\theta) := \frac{1}{s} \sum_{i=1}^{s} \nabla_{\theta}^2 \ell(\theta; x_i)$, and $\sup_{\theta \in \mathbb{B}(\hat{\theta}_{\star}, r_0)} \left\| (\nabla \mathcal{L}_s - \nabla \mathcal{L}_{ideal})(\theta) - (\nabla \mathcal{L}_s - \nabla \mathcal{L}_{ideal})(\hat{\theta}_{\star}) \right\| \leq \bar{c}\sqrt{\frac{\log(1/\delta)}{s}} \|\theta - \hat{\theta}_{\star}\|$. (A4) (ERM stays local) There exists a sequence $\delta_s \downarrow 0$ such that $\Pr\left(\hat{\theta}_s \in \mathbb{B}(\hat{\theta}_{\star}, r_0)\right) \geq 1 - \delta_s$. (A5) (Boundedness near optimum) There exists $B < \infty$ such that $\mathcal{L}_{ideal}(\theta) \leq B$ for all $\theta \in \mathbb{B}(\hat{\theta}_{\star}, r_0)$. See more details about assumptions in Appendix B.2.*

*Then there exist constants $C_1, C_2, C_b > 0$ depending only on $(\mu, L_H)$, such that for all $s$,*

$$\mathbb{E}\left[\mathcal{L}_{ideal}(\hat{\theta}_s) - \mathcal{L}_{ideal}(\hat{\theta}_{\star})\right] \geq \frac{1}{2s} \mathrm{tr}\left(H_{\star}^{-1} \Sigma_{\star}\right) - \frac{C_1}{s^{3/2}} - \frac{C_2}{s^2} - C_b \delta_s . \tag{3}$$

**Visualization of the Limitations of Limited Soft Label Supervision.** To illustrate the limitations of limited soft-label coverage, we compare the model's behavior on both the training and test sets, as shown on the right part of Fig. 2. Under finite soft-label supervision, the test-time loss landscape deviates notably from that of the training set, indicating overfitting and reduced generalization.

### 3.3 CALIBRATING LVSD WITH ACCURATE SUPERVISION

To mitigate *LVSD* arising under finite-$s$ soft label coverage, we propose **H**ard Label to **A**lleviate **L**ocal Semantic **D**rift (**HALD**), a *soft→hard→soft* calibration schedule. Intuitively, we first allow the student model to acquire coarse discriminative ability from finite-$s$ teacher soft labels, increasing gradient alignment between hard and soft supervision to ensure a smooth transition. We then *de-LVSD* the student by enforcing class-accurate constraints with hard labels, suppressing crop-specific variance. Finally, we resume teacher-guided learning to align the student with the teacher distribution on the variance-reduced representation, achieving a balance between reliance on limited soft labels and the global semantics provided by hard labels, thereby enhancing overall performance. We will then formally describe these three stages.

**How to determine the training duration for each stage.** We assume (as in our theoretical framework) that the model can fit the finite-$s$ soft-label supervision on $\Omega_{soft}$ to empirical risk minimization (ERM). Let $n_{soft}$ denote the epoch budget required for the model to reach convergence on $\Omega_{soft}$, and let $n_{total}$ be the total training budget. We allocate the remaining epochs to hard-label calibration: $n_{hard} := n_{total} - n_{soft} \ (\geq 0)$. Training then follows:

$$\underbrace{T_A = \left\lfloor \frac{n_{soft}}{2} \right\rfloor}_{soft}, \qquad \underbrace{T_B = n_{hard}}_{hard}, \qquad \underbrace{T_C = n_{soft} - T_A}_{soft},$$

If $n_{\text{total}} \leq n_{\text{soft}}$, we set $n_{\text{hard}} = 0$ and run soft-label training only. This schedule preserves the ERM fit on $\Omega_{\text{soft}}$, inserts a hard-label calibration phase of length $n_{\text{hard}}$ to mitigate local semantic drift, and finally re-aligns with $\Omega_{\text{soft}}$ to consolidate the variance reduction benefits.

*(i) Stage A (soft pretraining).* Let $s$ denote the *total* number of pre-generated soft labels (SLC $\times$ C). Define the global soft-label pool:

$$\Omega_{\text{soft}} := \left\{ (x_i^{(\text{crop})}, \tilde{p}_i) \right\}_{i=1}^s, \qquad \tilde{p}_i := \tilde{p}(\cdot \mid x_i^{(\text{crop})}),$$

where each $x_i^{(\text{crop})}$ is obtained by sampling a training image $\tilde{x}$ and a crop $x \sim \mathcal{T}(\tilde{x})$. At step $t$, sample indices $J_t = \{j_1, \ldots, j_B\} \subset \{1, \ldots, s\}$ *with replacement* and form the mini-batch $\{(x_{j_b}^{(\text{crop})}, \tilde{p}_{j_b})\}_{b=1}^B$. Using the same per-crop loss $\mathcal{L}[\cdot, \cdot]$, minimize the batch estimator,

$$\widehat{\mathcal{L}}_{\text{soft}}^{(t)}(\theta) = \frac{1}{B} \sum_{b=1}^B \mathcal{L}\big(\tilde{p}_{j_b}, q_\theta(\cdot \mid x_{j_b}^{(\text{crop})})\big), \qquad \theta_{t+1} = \theta_t - \eta_t \nabla_\theta \widehat{\mathcal{L}}_{\text{soft}}^{(t)}(\theta_t).$$

We denote by $\hat{\theta}_s^{\text{A}}$ the parameters obtained after Stage A training using this pool-sampling procedure.

*(ii) Stage B (de-LVSD via hard labels).* Define label smoothing and the CutMix target (for $C$ classes),

$$\text{LS}_\alpha(y) = (1 - \alpha)\,\delta_y + \alpha\,\tfrac{1}{C}, \qquad t_{\lambda,\alpha}(y, y') = (1 - \lambda)\,\text{LS}_\alpha(y) + \lambda\,\text{LS}_\alpha(y').$$

Let the sampling space be: $\Omega_{\text{cal}} := \Big\{ ((\tilde{x}, y), (\tilde{x}', y'), x, x', \lambda, m) : \; x \sim \mathcal{T}(\tilde{x}), \; x' \sim \mathcal{T}(\tilde{x}'), \; \lambda \in (0, 1), \; m \in \mathcal{M} \Big\}$. For any $\omega = ((\tilde{x}, y), (\tilde{x}', y'), x, x', \lambda, m) \in \Omega_{\text{cal}}$, define the calibration loss,

$$\ell_{\text{cal}}(\theta; \omega) := \mathcal{L}\Big( t_{\lambda,\alpha}(y, y'), q_\theta(\cdot \mid \text{CM}_{\lambda,m}(x, x')) \Big).$$

Initialize $\theta_0 := \hat{\theta}_s^{\text{A}}$. At each step $t$, draw an i.i.d. minibatch $\{\omega_i^{(t)}\}_{i=1}^B \subset \Omega_{\text{cal}}$ and update,

$$\widehat{\mathcal{L}}_{\text{cal}}^{(t)}(\theta) = \tfrac{1}{B} \sum_{i=1}^B \ell_{\text{cal}}(\theta; \omega_i^{(t)}), \quad \theta_{t+1} = \theta_t - \eta_t\,\nabla_\theta \widehat{\mathcal{L}}_{\text{cal}}^{(t)}(\theta_t).$$

As crops and CutMix geometry are resampled at every step, minibatches are effectively non-repeating and provide ground-truth–anchored, diverse local views of each base image, thereby mitigating the semantic bias induced by finite-$s$ soft-label supervision in Stage A.

*(iii) Stage C (soft refinement).* Initialize from $\hat{\theta}^{\text{B}}$; Stage C *follows Stage A's* pool-based protocol (same sampler on $\Omega_{\text{soft}}$ and per-crop loss $\mathcal{L}$), yielding final $\hat{\theta}$.

### 3.4 THEORETICAL ANALYSIS FOR **HALD**

**Optimization Coherence and Stability.** Theorem 3 shows that the alignment between soft- and hard-label gradients is controlled by the ratio $D/m_0$, where $D$ denotes the inter-class gradient spread and $m_0$ the minimal gradient norm. As training progresses under any form of supervision that allows the model to converge, the representation space gradually stabilizes and the classifier head becomes more aligned across samples. Consequently, $D$ tends to decrease faster than $m_0$, leading to a monotonically decreasing ratio $D/m_0$ and a progressively tighter alignment bound. Empirically, as shown in Fig. 3, the cosine similarity between soft- and hard-label gradients remains positive and increases steadily throughout training, confirming the theoretical prediction.

The *Soft–Hard–Soft* design naturally follows from this theory: the first Soft stage aligns the model and strengthens gradient similarity; the **Hard stage** reduces variance and corrects semantic drift; and the final stage restores fine-grained teacher consistency on the variance-reduced representation.

**Theorem 3** (Soft–Hard Gradient Consistency; proof in Appendix B.3.1). *Fix a crop $x^{(\text{crop})} \sim \mathcal{T}(\tilde{x})$ and $C$ classes. Let $g_c := \nabla_\theta \log q_\theta(c \mid x^{(\text{crop})})$, $\nabla_\theta \mathcal{L}_{\text{soft}} = -\sum_c \tilde{p}_c g_c$, $\nabla_\theta \mathcal{L}_{\text{hard}} = -\sum_c \bar{p}_c^{(\alpha)} g_c$ (where $\bar{p}^{(\alpha)}$ is the $\alpha$-smoothed one–hot). Assume $D := \sup_{i \neq j} \|g_i - g_j\| < \infty$ and $m_0 := \min\{\|\sum_c \tilde{p}_c g_c\|, \|\sum_c \bar{p}_c^{(\alpha)} g_c\|\} > 0$. Then there exists a constant $C_{\text{align}}(\tilde{x}, \alpha)$ depending only on the teacher's predictive entropy and the smoothing rate such that,*

$$\boxed{\mathbb{E}_{\text{crop}}[\cos(\nabla_\theta \mathcal{L}_{\text{soft}}, \nabla_\theta \mathcal{L}_{\text{hard}})] \; \geq \; 1 - \frac{D}{m_0} \cdot C_{\text{align}}(\tilde{x}, \alpha).}$$

**Analysis of Hard Label Calibration.** By Corollary 1, hard-label calibration increases the effective sample size from $s$ to at least $s_{\text{eff}}$, thereby improving both the optimization objective in Equations 2 and the generalization bound in Equation 3 via variance reduction. The performance gain is visualized in Fig. 2. *Intuitively*, hard-label calibration mitigates local semantic drift by enlarging the effective sample size, which reduces the sample variance and alleviates overfitting arising from finite-$s$ soft-label supervision. This improvement is driven by the strong alignment between soft- and hard-label gradients (high expected cosine similarity), ensuring that optimization on hard labels remains informative about unseen crops drawn from the same population distribution.

**Corollary 1** (Proof in Appendix B.4). *Assume the conditions of Theorem 3 hold so that $\mathbb{E}\big[\langle u,\,v\rangle\big] = \mathbb{E}\big[\cos\big(v_{\text{soft}}, v_{\text{hard}}\big)\big] \geq \rho_\star \in (0,1)$, where $u := v_{\text{soft}}/\|v_{\text{soft}}\|$ and $v := v_{\text{hard}}/\|v_{\text{hard}}\|$. Let $(u_i, v_i)_{i=1}^s$ be i.i.d. copies of $(u, v)$. Then the effective sample size satisfies:*

$$s_{\text{eff}} \geq \frac{s}{1 - \rho_\star^2}.$$

# 4 EXPERIMENT

## 4.1 EXPERIMENT SETTINGS

Additional details on datasets, distillation methods, and setup are provided in the Appendix.

**Datasets.** We evaluate **HALD** on Tiny-ImageNet ($64\times64$, $C{=}200$) (Le & Yang, 2015) and ImageNet-1K ($224\times224$, $C{=}1000$) (Deng et al., 2009), covering both low- and high-resolution regimes with distinct class scales for comprehensive evaluation.

**Generation Methods.** We consider four representative paradigms: (i) SRe$^2$L (Yin et al., 2023), an optimization-based method prone to limited diversity and distribution shift; (ii) LPLD (Xiao & He, 2024), a diversity-enhanced variant; (iii) RDED (Sun et al., 2024), real-data selection via class-preserving crops; and (iv) FADRM (Cui et al., 2025a), a residual-hybrid approach that fuses real-image priors with optimization. These span synthetic–real and low–high diversity axes.

**Baselines methods.** We compare the proposed *Soft–Hard–Soft* training paradigm (**HALD**) with three baselines: (1) *Soft-Only*, which relies solely on soft labels for supervision; (2) GIFT (Shang et al., 2024), which incorporates hard-label information directly into the soft labels during training; and (3) Joint Objective, which optimizes a combined loss that equally leverages both supervision sources: $\mathcal{L} = \mathcal{L}_{\text{soft}} + \lambda \mathcal{L}_{\text{hard}}$, where $\lambda$ controls the relative weighting between the soft- and hard-label objectives. Unless otherwise specified, all generation techniques adopt the strongest baseline, namely the *Soft-Only* protocol, whereas our method employs FADRM for generation and **HALD** for training. Results are reported as Top-1 accuracy (%) with mean $\pm$ std over three runs, and table-specific settings (IPC, SLC, architecture, dataset) are detailed alongside each result.

## 4.2 MAIN RESULT

As shown in Table 1 and Table 2, our method consistently achieves superior performance. With SLC= 250 and a 50-IPC distilled dataset, **HALD** reaches 47.6% Top-1 accuracy on ImageNet-1K, surpassing the previous SOTA LPLD by +13.4%, thereby validating its effectiveness.

Table 1: Comparison with SOTA methods on Tiny-ImageNet.

| | SLI = 2 | | | | | SLI = 1 | | | | |
|---|---|---|---|---|---|---|---|---|---|---|
| | SRe$^2$L | RDED | FADRM | LPLD | Ours | SRe$^2$L | RDED | FADRM | LPLD | Ours |
| IPC=10 | 14.6$\pm$0.2 | 12.5$\pm$0.3 | 17.4$\pm$0.5 | 13.3$\pm$0.3 | **22.8**$\pm$0.3 | 8.3$\pm$0.4 | 7.7$\pm$0.2 | 10.1$\pm$0.3 | 8.2$\pm$0.2 | **18.6**$\pm$0.4 |
| *Storage* | SLC = 20 (1.52 MB) | | | | | SLC = 10 (0.76 MB) | | | | |
| IPC=20 | 21.8$\pm$0.3 | 19.6$\pm$0.4 | 26.4$\pm$0.2 | 21.3$\pm$0.4 | **29.7**$\pm$0.5 | 14.8$\pm$0.3 | 11.7$\pm$0.2 | 17.5$\pm$0.2 | 14.0$\pm$0.3 | **25.9**$\pm$0.3 |
| *Storage* | SLC = 40 (3.04 MB) | | | | | SLC = 20 (1.52 MB) | | | | |
| IPC=30 | 27.3$\pm$0.5 | 23.6$\pm$0.6 | 31.0$\pm$0.4 | 27.5$\pm$0.5 | **33.8**$\pm$0.4 | 19.7$\pm$0.2 | 17.9$\pm$0.5 | 23.8$\pm$0.4 | 17.4$\pm$0.4 | **28.7**$\pm$0.3 |
| *Storage* | SLC = 60 (4.56 MB) | | | | | SLC = 30 (2.28 MB) | | | | |
| IPC=40 | 29.6$\pm$0.2 | 26.8$\pm$0.5 | 32.9$\pm$0.4 | 29.1$\pm$0.4 | **35.2**$\pm$0.3 | 22.2$\pm$0.3 | 19.6$\pm$0.6 | 26.6$\pm$0.3 | 22.1$\pm$0.5 | **30.3**$\pm$0.4 |
| *Storage* | SLC = 80 (6.08 MB) | | | | | SLC = 40 (3.04 MB) | | | | |
| IPC=50 | 31.9$\pm$0.2 | 27.9$\pm$0.4 | 36.0$\pm$0.3 | 34.3$\pm$0.3 | **38.2**$\pm$0.5 | 24.0$\pm$0.4 | 20.5$\pm$0.2 | 27.8$\pm$0.5 | 24.1$\pm$0.2 | **30.7**$\pm$0.4 |
| *Storage* | SLC = 100 (7.60 MB) | | | | | SLC = 50 (3.80 MB) | | | | |

Table 2: Comparison with SOTA methods on ImageNet-1K. $^\dagger$ denotes the reported results.

| | SLI = 10 | | | | | SLI = 5 | | | | |
|---|---|---|---|---|---|---|---|---|---|---|
| | SRe$^2$L | RDED | FADRM | LPLD | Ours | SRe$^2$L | RDED | FADRM | LPLD | Ours |
| IPC=10 | 18.6±0.3 | 18.1±0.3 | 26.5±0.2 | 23.1$^\dagger$±0.1 | **37.0**±0.5 | 8.6±0.5 | 8.4±0.3 | 11.1±0.6 | 8.1±0.4 | **27.0**±0.6 |
| *Storage* | SLC = 100 (190 MB) | | | | | SLC = 50 (95 MB) | | | | |
| IPC=20 | 29.3±0.4 | 23.9±0.2 | 34.7±0.4 | 35.9$^\dagger$±0.3 | **43.1**±0.4 | 19.4±0.3 | 15.7±0.5 | 23.6±0.5 | 24.0±0.3 | **39.2**±0.4 |
| *Storage* | SLC = 200 (380 MB) | | | | | SLC = 100 (190 MB) | | | | |
| IPC=30 | 34.3±0.2 | 30.6±0.5 | 41.1±0.6 | 38.0±0.4 | **47.7**±0.2 | 23.7±0.6 | 20.8±0.3 | 28.7±0.3 | 26.9±0.3 | **43.6**±0.5 |
| *Storage* | SLC = 300 (570 MB) | | | | | SLC = 150 (285 MB) | | | | |
| IPC=40 | 43.3±0.2 | 40.9±0.4 | 50.0±0.5 | 44.9±0.3 | **52.6**±0.3 | 28.8±0.3 | 24.1±0.6 | 33.1±0.6 | 30.2±0.5 | **45.6**±0.3 |
| *Storage* | SLC = 400 (760 MB) | | | | | SLC = 200 (380 MB) | | | | |
| IPC=50 | 46.8±0.3 | 43.5±0.4 | 52.7±0.2 | 47.2±0.3 | **53.7**±0.2 | 33.9±0.4 | 29.2±0.6 | 39.3±0.4 | 34.2±0.3 | **47.6**±0.5 |
| *Storage* | SLC = 500 (950 MB) | | | | | SLC = 250 (475 MB) | | | | |

Table 3: **Left:** Impact of soft-label phase length on final performance (all generation methods are trained with **HALD**). **Right:** Results on cross-architecture generalization, showing Top-1 accuracy (%) with IPC=10 under SLC=100 on different neural networks.

| Method | Soft-Label Phase Length (epochs) | | | |
|---|---|---|---|---|
| | 100 | 150 | **200** | 250 |
| FADRM | 31.3 | 34.8 | **37.0** | 29.3 |
| RDED | 16.6 | 24.0 | **25.4** | 22.5 |
| LPLD | 24.1 | 27.1 | **28.8** | 26.3 |
| SRe$^2$L | 26.7 | 30.9 | **31.7** | 26.4 |

*Soft-label convergence length = 200 epochs*

| Model | #Params | RDED | LPLD | FADRM | **Ours** |
|---|---|---|---|---|---|
| ResNet-18 | 11.7M | 18.1 | 23.1 | 26.5 | **37.0** $^{\uparrow 10.5}$ |
| ResNet-50 | 25.6M | 25.2 | 27.3 | 36.1 | **38.0** $^{\uparrow 1.9}$ |
| EfficientNet-B0 | 39.6M | 19.5 | 26.1 | 36.4 | **37.8** $^{\uparrow 1.4}$ |
| MobileNetV2 | 3.4M | 17.3 | 25.1 | 34.2 | **35.3** $^{\uparrow 1.1}$ |
| DenseNet121 | 8.0M | 28.3 | 36.7 | 43.3 | **44.3** $^{\uparrow 1.0}$ |
| ShuffleNetV2-0.5x | 1.4M | 17.9 | 21.1 | 29.2 | **32.3** $^{\uparrow 3.1}$ |
| Vit-Tiny | 13M | 3.2 | 3.8 | 5.6 | **8.9** $^{\uparrow 3.3}$ |
| VGG-11 | 133M | 26.3 | 28.9 | 31.0 | **33.6** $^{\uparrow 2.6}$ |
| VGG-16 | 138M | 28.9 | 34.3 | 36.2 | **37.4** $^{\uparrow 1.2}$ |

**Comparison with more baselines.** As summarized in Table 4, HALD achieves the best overall performance, while GIFT performs comparably to the soft-only baseline. For the Joint Objective, performance declines as $\lambda$ increases and peaks at $\lambda = 0$ (soft-only), indicating that jointly mixing hard and soft supervision introduces gradient inconsistency that degrades performance. In contrast, HALD's stage-wise design leverages their alignment sequentially, mitigating this conflict and yielding consistent gains.

Table 4: Performance comparison under identical storage and training budgets, highlighting HALD's advantage through stage-wise soft–hard integration. JO denotes the Joint Objective method.

| SLC | GIFT | JO. $\lambda = 1$ | JO. $\lambda = 0.1$ | JO $\lambda = 0.01$ | Soft Only | Ours |
|---|---|---|---|---|---|---|
| 100 | 27.0 | 5.9 | 7.0 | 13.1 | 26.9 | **43.5** |
| 200 | 39.1 | 8.1 | 9.4 | 17.5 | 39.2 | **47.3** |
| 300 | 46.7 | 9.5 | 10.3 | 20.1 | 46.6 | **50.7** |

**Storage Efficiency.** Table 5 reports performance under varying soft label storage budgets. While LPLD degrades sharply with tighter constraints, **HALD** maintains strong accuracy (e.g., 35.3% at 95M, +27.2%). This demonstrates the storage efficiency of our calibration strategy, which effectively enhances the utility of stored soft labels under limited capacity.

Table 5: Storage *vs.* Effectiveness. (ImageNet-1K IPC=10)

| | 571M | 476M | 381M | 285M | 190M | 95M |
|---|---|---|---|---|---|---|
| LPLD | 43.1 | 34.2 | 34.7 | 33.7 | 24.6 | 8.1 |
| **Ours** | 46.9 $^{\uparrow 3.8}$ | 47.6 $^{\uparrow 13.4}$ | 46.3 $^{\uparrow 11.6}$ | 41.8 $^{\uparrow 8.1}$ | 41.9 $^{\uparrow 17.3}$ | 35.3 $^{\uparrow 27.2}$ |

## 4.3 ANALYSIS

**LVSD Quantification.** To quantify the degree of LVSD across teacher models, we define the *LVSD ratio* as $R(\tilde{x}) = \frac{\mathrm{Tr}(\hat{\Sigma}_{\text{strong}})}{\mathrm{Tr}(\hat{\Sigma}_{\text{weak}})+\varepsilon}$, where the numerator measures the prediction variance under *strong augmentations* that correspond to local views of the image (e.g., aggressive random resized cropping), and the denominator under *weak augmentations* that preserve global semantics (resize and center crop). Hence, $R(\tilde{x})$ captures the degree of semantic drift between local and global views.

Table 6: Quantitative analysis of Local-View Semantic Drift (LVSD) across teacher models. A larger $R$ indicates higher prediction variance under local views relative to gloal views, confirming that LVSD is substantial across architectures.

| Teacher | Mean $\mathrm{Tr}(\hat{\Sigma}_{\text{weak}})$ | Mean $\mathrm{Tr}(\hat{\Sigma}_{\text{strong}})$ | $\log_{10}(\text{Mean } R)$ | $\mathbf{p}(R > 1)$ |
|---|---|---|---|---|
| ResNet-18 | $4.84 \times 10^{-15}$ | 0.0102 | 3.27 | 97.2% |
| MobileNetV2 | $4.42 \times 10^{-15}$ | 0.3756 | 5.28 | 99.2% |
| ShuffleNetV2 | $4.51 \times 10^{-15}$ | 0.0591 | 5.35 | 98.0% |

Table 7: Comprehensive ablation of the impact of incorporating hard-label supervision across state-of-the-art dataset distillation methods on ImageNet-1K and Tiny-ImageNet. All models are trained for 300 epochs under identical hyperparameters, with the evaluation protocol being the sole difference. $^\dagger$ denotes values reported by the corresponding original sources.

| IPC | Generation | Evaluation | ImageNet-1K | | | | | | Tiny-ImageNet | |
|---|---|---|---|---|---|---|---|---|---|---|
| | | | SLC=300 | SLC=250 | SLC=200 | SLC=150 | SLC=100 | SLC=50 | SLC=100 | SLC=50 |
| IPC=10 | SRe$^2$L | Soft-Only | 32.2 | 29.9 | 28.0 | 24.3 | 18.6 | 8.6 | 31.4 | 24.0 |
| | | **Ours** | 35.2 $^{\uparrow 3.0}$ | 35.9 $^{\uparrow 6.0}$ | 33.3 $^{\uparrow 5.3}$ | 31.3 $^{\uparrow 7.0}$ | 31.7 $^{\uparrow 13.1}$ | 23.8 $^{\uparrow 15.2}$ | 31.9 $^{\uparrow 0.5}$ | 25.5 $^{\uparrow 1.5}$ |
| | RDED | Soft-Only | 26.2 | 24.1 | 21.4 | 18.8 | 18.1 | 8.3 | 27.6 | 22.3 |
| | | **Ours** | 27.2 $^{\uparrow 1.0}$ | 26.2 $^{\uparrow 2.1}$ | 25.5 $^{\uparrow 4.1}$ | 22.8 $^{\uparrow 4.0}$ | 25.4 $^{\uparrow 7.3}$ | 16.9 $^{\uparrow 8.6}$ | 31.0 $^{\uparrow 3.4}$ | 27.0 $^{\uparrow 4.7}$ |
| | LPLD | Soft-Only | 32.7 $^\dagger$ | 34.9 | 32.7 | 28.6$^\dagger$ | 23.1$^\dagger$ | 8.1 | 31.5 | 23.5 |
| | | **Ours** | 37.0 $^{\uparrow 4.3}$ | 36.7 $^{\uparrow 1.8}$ | 33.9 $^{\uparrow 1.2}$ | 31.9 $^{\uparrow 3.3}$ | 28.8 $^{\uparrow 5.7}$ | 20.5 $^{\uparrow 12.4}$ | 32.6 $^{\uparrow 1.1}$ | 26.5 $^{\uparrow 3.0}$ |
| | FADRM | Soft-Only | 42.1 | 40.7 | 39.0 | 35.3 | 26.5 | 11.1 | 34.4 | 28.1 |
| | | **Ours** | 43.4 $^{\uparrow 1.3}$ | 42.2 $^{\uparrow 1.5}$ | 40.4 $^{\uparrow 1.4}$ | 38.6 $^{\uparrow 3.3}$ | 37.0 $^{\uparrow 10.5}$ | 27.0 $^{\uparrow 15.9}$ | 36.2 $^{\uparrow 1.8}$ | 30.7 $^{\uparrow 2.6}$ |
| IPC=20 | SRe$^2$L | Soft-Only | 35.1 | 30.6 | 29.3 | 21.8 | 19.4 | 6.8 | 30.9 | 22.0 |
| | | **Ours** | 40.5 $^{\uparrow 5.4}$ | 38.0 $^{\uparrow 7.4}$ | 36.2 $^{\uparrow 6.9}$ | 35.5 $^{\uparrow 13.7}$ | 31.6 $^{\uparrow 12.2}$ | 21.7 $^{\uparrow 14.9}$ | 32.7 $^{\uparrow 1.8}$ | 24.5 $^{\uparrow 2.5}$ |
| | RDED | Soft-Only | 29.2 | 26.4 | 23.9 | 20.2 | 15.7 | 7.1 | 30.1 | 20.7 |
| | | **Ours** | 35.5 $^{\uparrow 6.3}$ | 33.9 $^{\uparrow 7.5}$ | 31.6 $^{\uparrow 7.7}$ | 29.5 $^{\uparrow 9.3}$ | 27.9 $^{\uparrow 12.2}$ | 19.1 $^{\uparrow 12.0}$ | 33.0 $^{\uparrow 2.9}$ | 25.2 $^{\uparrow 4.5}$ |
| | LPLD | Soft-Only | 41.0$^\dagger$ | 38.5 | 35.9$^\dagger$ | 33.0$^\dagger$ | 24.0 | 7.8 | 35.2 | 21.2 |
| | | **Ours** | 42.8 $^{\uparrow 1.8}$ | 40.9 $^{\uparrow 2.4}$ | 40.6 $^{\uparrow 4.7}$ | 37.2 $^{\uparrow 4.2}$ | 34.9 $^{\uparrow 10.9}$ | 20.9 $^{\uparrow 13.1}$ | 36.0 $^{\uparrow 0.8}$ | 24.6 $^{\uparrow 3.4}$ |
| | FADRM | Soft-Only | 39.9 | 36.8 | 34.7 | 30.0 | 23.6 | 8.4 | 37.0 | 26.8 |
| | | **Ours** | 46.4 $^{\uparrow 6.5}$ | 44.8 $^{\uparrow 8.0}$ | 43.1 $^{\uparrow 8.4}$ | 40.9 $^{\uparrow 10.9}$ | 39.2 $^{\uparrow 15.6}$ | 27.1 $^{\uparrow 18.7}$ | 37.6 $^{\uparrow 0.6}$ | 29.8 $^{\uparrow 3.0}$ |
| IPC=50 | SRe$^2$L | Soft-Only | 36.3 | 33.9 | 30.5 | 24.6 | 23.9 | 8.6 | 31.9 | 24.0 |
| | | **Ours** | 41.2 $^{\uparrow 4.9}$ | 40.8 $^{\uparrow 6.9}$ | 38.1 $^{\uparrow 7.6}$ | 33.6 $^{\uparrow 9.0}$ | 31.3 $^{\uparrow 7.4}$ | 27.9 $^{\uparrow 19.3}$ | 32.9 $^{\uparrow 1.0}$ | 26.0 $^{\uparrow 2.0}$ |
| | RDED | Soft-Only | 31.5 | 29.2 | 25.7 | 20.7 | 21.0 | 12.7 | 27.9 | 20.5 |
| | | **Ours** | 42.3 $^{\uparrow 10.8}$ | 39.1 $^{\uparrow 9.9}$ | 38.7 $^{\uparrow 13.0}$ | 38.1 $^{\uparrow 17.4}$ | 38.5 $^{\uparrow 17.5}$ | 32.8 $^{\uparrow 20.1}$ | 31.1 $^{\uparrow 3.2}$ | 26.4 $^{\uparrow 5.9}$ |
| | LPLD | Soft-Only | 43.1$^\dagger$ | 34.2 | 34.7 | 33.7$^\dagger$ | 24.6 | 8.1 | 34.4 | 24.1 |
| | | **Ours** | 44.8 $^{\uparrow 1.7}$ | 41.1 $^{\uparrow 6.9}$ | 39.8 $^{\uparrow 5.1}$ | 37.3 $^{\uparrow 3.6}$ | 32.1 $^{\uparrow 7.5}$ | 26.5 $^{\uparrow 18.4}$ | 36.3 $^{\uparrow 1.9}$ | 27.8 $^{\uparrow 3.7}$ |
| | FADRM | Soft-Only | 42.3 | 39.3 | 43.6 | 30.4 | 30.6 | 18.3 | 36.0 | 27.8 |
| | | **Ours** | 46.9 $^{\uparrow 4.6}$ | 47.6 $^{\uparrow 8.3}$ | 46.3 $^{\uparrow 2.7}$ | 41.8 $^{\uparrow 11.4}$ | 41.9 $^{\uparrow 11.3}$ | 35.3 $^{\uparrow 17.0}$ | 38.2 $^{\uparrow 2.2}$ | 30.7 $^{\uparrow 2.9}$ |

As shown in Table 6, $R$ is markedly large across backbones, indicating that LVSD is consistently severe, thereby motivating the need for semantic calibration under limited soft label coverage.

**Semantic Calibration.** To verify that HALD mitigates semantic drift and improve generalization, we analyze both crop-level consistency and prediction alignment with a reference model trained under full soft-label coverage. Specifically, crop-level consistency quantifies how well predictions from different crops of the same image agree, measured by the average Jensen–Shannon (JS) divergence and cosine similarity before and after Stage B. Prediction alignment, on the other hand, evaluates how closely the student model's predictions (w/ or w/o hard Calibration) match those of the reference model on unseen data. As shown in Table 8, we observed improved semantic consistency and stronger prediction alignment with the reference model, validating the role of hard labels in mitigating semantic drift and improving overall performance.

Table 8: **Left:** Crop-level consistency before and after Stage B. **Right:** Prediction alignment with a reference model trained under full soft-label coverage on unseen data.

| Stage | JS Div. | Cos. Sim. | | JS Div. | Cos. Sim. |
|---|---|---|---|---|---|
| Before Stage B | 0.1811 | 0.744 | w/o Hard Calibration | 0.337 | 0.458 |
| After Stage B | **0.0393** | **0.959** | w/ Hard Calibration | **0.226** | **0.623** |

## 4.4 Cross-Architecture Generalization

To assess backbone-agnostic effectiveness, we evaluate **HALD** across heterogeneous backbones, ranging from lightweight networks to larger architectures under same hyper-parameters. As reported in the right of Table 3, **HALD** yields consistent improvements across all backbones examined. For instance, **HALD** improves ShuffleNetV2 by +3.1%. These results indicate that the benefits of our training paradigm are architecture-agnostic and scale with parameter counts and model capacities.

## 4.5 Ablation

**Impact of Hard-Label Calibration.** To assess the generality of **HALD**, we compare the *Soft–Hard–Soft* (ours) schedule with *Soft–Only* across multiple dataset distillation techniques (Table 7). Consistent gains across all methods confirm the benefit of hard-label calibration, especially under low-SLC settings where local-view semantic drift is more pronounced.

**When to Switch to Hard Labels.** To evaluate the effectiveness of our proposed training paradigm, we train **HALD** under four schedules. As reported in Table 9, *Soft–Hard–Soft* achieves the highest accuracy across settings, whereas the remaining schedules underperform. These results show that introducing hard labels mid-training is most effective, aligning with Theorem 3, which predicts stronger gradient alignment after partial training.

Table 9: Impact of different training schedules under SLC=100 on an IPC=10 distilled FADRM dataset.

| Hard-Soft | Soft-Hard | Hard-Soft-Hard | Soft-Hard-Soft |
|---|---|---|---|
| 17.0 % | 14.2 % | 11.3 % | **37.0** % |

**How Long to Use Hard Labels.** We validate our theoretical assumption that the soft-label phase should match the convergence time of standalone soft-label training. As shown in the left of Table 3, extending this phase to its full predefined length consistently improves performance.

Table 10: Effect of Label-Smoothing Rate (All methods use **HALD**).

|  | 0.0 | 0.2 | 0.4 | 0.6 | 0.8 | 0.9 |
|---|---|---|---|---|---|---|
| FADRM | 35.3 | 35.5 | 35.7 | 35.9 | **37.0** | 36.3 |
| LPLD | 26.9 | 27.3 | 27.5 | 27.9 | **28.8** | 27.9 |
| SRe$^2$L | 29.6 | 30.0 | 30.9 | 31.5 | **31.7** | 31.3 |
| RDED | 21.9 | 22.9 | 23.8 | 24.0 | **25.4** | 24.5 |

**Effect of Label-Smoothing ($\alpha$).** As shown in Table 10, the optimal $\alpha$ is 0.8 for both generation techniques, likely due to the use of high-temperature soft label training, where a larger $\alpha$ better preserves the corresponding high-entropy label distribution.

**Effect of the first and last soft-label stages.** As shown in Table 11, both the first and last soft-label phases are essential, as allocating more training budget to either phase leads to inferior performance compared with the balanced setting. When the budget is biased toward the first phase, the model after semantic calibration lacks sufficient samples to relearn the teacher's fine-grained semantics. Conversely, emphasizing the last phase causes the model to enter the hard-label stage before full convergence, resulting in poor gradient alignment and weakened semantic calibration. Therefore, we allocate the soft-label duration equally between the two phases.

Table 11: Effect of first and last soft-label phase durations on performance.

| First Soft Duration | Last Soft Duration | HALD |
|---|---|---|
| 50 | 100 | 35.2 |
| 100 | 50 | 34.7 |
| 75 | 75 | **35.6** |

## 4.6 Conventional Large-Scale Dataset Experiment

To assess generalization beyond synthetic data, we evaluate **HALD** on a randomly sampled subset from original ImageNet-1K. As shown in Table 12, incorporating hard-label supervision consistently improves performance, confirming the effectiveness of **HALD** on real data.

Table 12: *Soft-Only* vs. **HALD** on real dataset.

| ResNet-18 | SLC=100 (190 MB) | | SLC=50 (95 MB) | |
|---|---|---|---|---|
| | *Soft-Only* | **Ours** | *Soft-Only* | **Ours** |
| IPC=10 | 29.9 | **34.4** | 26.9 | **28.6** |
| IPC=50 | 26.9 | **44.7** | 19.1 | **40.9** |

## 5 Conclusion

In this work, we revisited the limits of soft-label supervision in dataset distillation under tight storage budgets and identified *local semantic drift* as a core failure mode when only a few per-image crops (and thus soft labels) are retained. We showed theoretically that the expected objective mismatch between reduced-crop and sufficient-crop training admits a strictly positive lower bound that scales inversely with the number of crops, and we proved that combining soft and hard labels does not introduce gradient inconsistency. Building on these insights, we proposed a lightweight calibration paradigm **HALD**, where hard labels act as content-agnostic anchors that realign supervision while preserving the fine-grained benefits of soft labels. Experiments on large-scale settings (e.g., ImageNet-1K) demonstrate that **HALD** mitigates drift, improves generalization and robustness, and substantially reduces storage overhead, providing a practical path toward scalable distillation.

## Ethics Statement

This work focuses on supervision design and storage efficiency, it neither collects new human subjects data nor accesses sensitive attributes beyond standard benchmarks. To minimize risks of original data leakage in the post-training phase, we explicitly isolate the teacher in post-training and avoid contact with original raw data, respecting practices in dataset distillation settings. We provide synthetic images and monitor class-wise disparities to reduce potential bias. Given the environmental

impact of the efficient training, we favor storage/compute-efficient protocols and disclose approximate storage savings. The distilled data is intended for research, any downstream development should follow local regulations and dataset licenses and avoid harmful or deceptive applications.

## REPRODUCIBILITY STATEMENT

All experiments are conducted on publicly available datasets, such as Tiny-ImageNet and ImageNet-1K. To ensure reproducibility, we fix random seeds for all stochastic components, and provide full details of hyper-parameters, training schedules, and model configurations in Appendix E. In addition, all experimental settings are managed via structured `.yml` configuration files, enabling modular and transparent control over the pipeline. The complete source code, along with configuration files and scripts for data preparation, training, and evaluation will be released.

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

# Appendix of HALD

CONTENTS

## A  NOTATION

| Symbol | Definition |
|---|---|
| $\mathcal{O}$ | Original dataset of labeled samples |
| $\mathcal{C}$ | Distilled dataset (small synthetic set) |
| $(x, y)$ | Input sample $x$ with ground-truth label $y$ |
| $\tilde{x}, \tilde{y}$ | Synthetic (distilled) sample and label |
| $f_\theta$ | Model parameterized by $\theta$ |
| $\theta_\mathcal{O}, \theta_\mathcal{C}$ | Parameters trained on $\mathcal{O}$ or $\mathcal{C}$ |
| $\mathcal{L}(\cdot, \cdot)$ | Per-sample loss functional |
| $\mathcal{T}(\tilde{x})$ | Augmentation distribution of $\tilde{x}$ |
| $x^{(\mathrm{crop})}$ | Random crop sampled from $\mathcal{T}(\tilde{x})$ |
| $\tilde{p}(x^{(\mathrm{crop})})$ | Teacher soft prediction on a crop |
| $\bar{p}$ | Crop-averaged teacher prediction $\mathbb{E}[\tilde{p}(x^{(\mathrm{crop})})]$ |
| $\Sigma$ | Covariance of teacher predictions across crops |
| $\hat{p}_s$ | Empirical average prediction over $s$ crops |
| SLC | Soft Labels per Class (storage budget) |
| $n_{\mathrm{soft}}, n_{\mathrm{hard}}, n_{\mathrm{total}}$ | Epoch budgets for soft-/hard-label training |
| $\Omega_{\mathrm{soft}}$ | Global pool of stored soft labels |
| $\Omega_{\mathrm{cal}}$ | Calibration sampling space with hard labels |
| $\mathrm{LS}_\alpha(\cdot)$ | Label smoothing distribution with ratio $\alpha$ |
| $t_{\lambda, \alpha}(y, y')$ | CutMix target between labels $y, y'$ |
| $q_\theta(\cdot \mid x)$ | Student predictive distribution on input $x$ |
| $\hat{\theta}_s^{\mathrm{A}}, \hat{\theta}^{\mathrm{B}}, \hat{\theta}$ | Parameters after Stages A, B, and C |
| $H_\star$ | Hessian of $\mathcal{L}_{\mathrm{ideal}}$ at optimum |
| $\Sigma_\star$ | Gradient covariance at optimum |
| $s_{\mathrm{eff}}$ | Effective sample size after calibration |

Table 13: List of common mathematical symbols used in this paper.

## B  PROOF

### B.1  LOWER BOUND ON THE EMPIRICAL LOSS BIAS UNDER LIMITED CROP SUPERVISION

*Proof of Theorem 1.* Fix the synthetic image $\tilde{x}$ and its augmentation law $\mathcal{T}(\tilde{x})$. Define the per-crop loss random variable:

$$X := \mathcal{L}\Big[\tilde{p}, q_\theta(\cdot \mid \tilde{x}^{(\mathrm{crop})})\Big], \quad \text{where } \tilde{x}^{(\mathrm{crop})} \sim \mathcal{T}(\tilde{x}).$$

Let $\mu := \mathbb{E}X = \mathcal{L}_{\mathrm{ideal}}(\theta; \tilde{x})$, $\sigma^2 := \mathrm{Var}(X) < \infty$, and a finite kurtosis $\kappa := \frac{\mathbb{E}\big[(X - \mu)^4\big]}{\sigma^4} \in [1, \infty)$. For $s$ i.i.d. crops $(\tilde{x}_i^{(\mathrm{crop})})_{i=1}^s$ drawn from $\mathcal{T}(\tilde{x})$, set:

$$X_i := \mathcal{L}\Big[\tilde{p}_i, q_\theta(\cdot \mid \tilde{x}_i^{(\mathrm{crop})})\Big] \quad \text{(i.i.d. copies of } X\text{)}, \qquad \bar{X}_s := \frac{1}{s}\sum_{i=1}^s X_i = \mathcal{L}_s(\theta; \tilde{x}).$$

Our target quantity is $\mathbb{E}\big[\,|\bar{X}_s - \mu|\,\big] = \mathbb{E}\big[\,|\mathcal{L}_s(\theta; \tilde{x}) - \mathcal{L}_{\mathrm{ideal}}(\theta; \tilde{x})|\,\big]$.

Step 1 (Second and fourth moments of the centered sample mean). Let $Y_i := X_i - \mu$ so that $\mathbb{E}Y_i = 0$, $\mathbb{E}Y_i^2 = \sigma^2$, and $\mathbb{E}Y_i^4 = \kappa\,\sigma^4$. Define the centered sample mean:

$$W := \bar{X}_s - \mu = \frac{1}{s}\sum_{i=1}^s Y_i.$$

By independence,

$$\mathbb{E}[W^2] = \frac{1}{s^2}\sum_{i=1}^s \mathbb{E}[Y_i^2] = \frac{\sigma^2}{s}.$$

For the fourth moment, only index patterns that are all equal or pairwise-equal contribute:

$$\mathbb{E}\left[\left(\sum_{i=1}^{s} Y_i\right)^4\right] \;=\; s\,\mathbb{E}[Y_1^4] \;+\; 3\,s(s-1)\,\sigma^4 \;=\; s\,\kappa\,\sigma^4 \;+\; 3s(s-1)\sigma^4.$$

Therefore,

$$\mathbb{E}[W^4] \;=\; \frac{1}{s^4}\,\mathbb{E}\left[\left(\sum_{i=1}^{s} Y_i\right)^4\right] \;=\; \frac{\sigma^4}{s^3}\left(\kappa + 3(s-1)\right).$$

**Step 2 (Paley–Zygmund on $W^2$).** Let $Z := W^2 \geq 0$. For any $\theta \in (0,1)$, Paley–Zygmund yields:

$$\mathbb{P}(Z \geq \theta\,\mathbb{E}Z) \;\geq\; (1-\theta)^2\,\frac{(\mathbb{E}Z)^2}{\mathbb{E}[Z^2]}.$$

Using $\mathbb{E}Z = \mathbb{E}[W^2] = \sigma^2/s$ and $\mathbb{E}[Z^2] = \mathbb{E}[W^4] = \frac{\sigma^4}{s^3}\left(\kappa + 3(s-1)\right)$ from Step 1,

$$\mathbb{P}\left(W^2 \geq \theta\,\frac{\sigma^2}{s}\right) \;\geq\; (1-\theta)^2\,\frac{\left(\frac{\sigma^2}{s}\right)^2}{\frac{\sigma^4}{s^3}\left(\kappa + 3(s-1)\right)} \;=\; (1-\theta)^2 \cdot \frac{s}{\kappa + 3(s-1)}.$$

**Step 3 (From a small-ball event to a first-moment bound).** For any $t > 0$, $\mathbb{E}|W| \geq t\,\mathbb{P}(|W| \geq t)$. Choose $t := \sqrt{\theta\,\mathbb{E}W^2} = \frac{\sigma}{\sqrt{s}}\sqrt{\theta}$ to match Step 2. Then:

$$\mathbb{E}|W| \;\geq\; \frac{\sigma}{\sqrt{s}}\sqrt{\theta}\,\mathbb{P}\left(W^2 \geq \theta\,\frac{\sigma^2}{s}\right) \;\geq\; \frac{\sigma}{\sqrt{s}}\sqrt{\theta}\,(1-\theta)^2\,\frac{s}{\kappa + 3(s-1)}.$$

**Step 4 (Optimize $\theta$).** Define $g(\theta) := \sqrt{\theta}\,(1-\theta)^2$ on $\theta \in [0,1]$. A direct derivative check gives $g'(\theta) = 0$ at $\theta^\star = \frac{1}{5}$ and $g(\theta^\star) = \frac{16}{25\sqrt{5}}$. Plugging $\theta^\star$ into Step 3 yields

$$\mathbb{E}|W| \;\geq\; \frac{\sigma}{\sqrt{s}} \cdot \frac{16}{25\sqrt{5}} \cdot \frac{s}{\kappa + 3(s-1)}.$$

**Step 5 (Uniform-in-$s$ simplification).** Consider $h(s) := \dfrac{s}{\kappa + 3(s-1)} = \dfrac{s}{3s + \kappa - 3}$ for $s \geq 1$. Then

$$h'(s) = \frac{(\kappa - 3)}{(\kappa + 3s - 3)^2}.$$

Hence:

- If $\kappa \geq 3$, $h$ is nondecreasing on $[1, \infty)$, so $\min_{s \geq 1} h(s) = h(1) = \frac{1}{\kappa} \leq \frac{1}{3}$.

- If $\kappa < 3$, $h$ is strictly decreasing and $\inf_{s \geq 1} h(s) = \lim_{s \to \infty} h(s) = \frac{1}{3}$, while $h(1) = \frac{1}{\kappa} > \frac{1}{3}$.

In both cases,

$$\frac{s}{\kappa + 3(s-1)} \;\geq\; \min\left\{\frac{1}{\kappa}, \frac{1}{3}\right\}.$$

Combining with the bound above concludes

$$\mathbb{E}\big[\big|\bar{X}_s - \mu\big|\big] \;=\; \mathbb{E}\big[\big|\mathcal{L}_s(\theta; \tilde{x}) - \mathcal{L}_{\text{ideal}}(\theta; \tilde{x})\big|\big] \;\geq\; \frac{\sigma}{\sqrt{s}} \cdot \frac{16}{25\sqrt{5}} \cdot \min\left\{\frac{1}{\kappa}, \frac{1}{3}\right\}.$$

This is exactly the claimed bound. $\qquad\qquad\qquad\qquad\qquad\qquad\qquad\qquad\qquad\qquad\qquad\qquad\square$

## B.2 Limited Crop Supervision Degrades Generalization Performance

**Interpretation of Assumptions (A1–A5).** To establish a finite-sample lower bound on the excess population risk of empirical risk minimization (ERM), we adopt five standard assumptions that ensure local regularity and statistical stability near the population minimizer $\hat{\theta}_\star$. Below we provide an intuitive interpretation of each:

- **(A1) Unbiased Score and Covariance.** We assume $\mathbb{E}[g(\hat{\theta}_\star; x)] = 0$ and that the covariance $\Sigma_\star = \mathrm{Cov}(g(\hat{\theta}_\star; x))$ exists with bounded $(2 + \kappa)$-moment. This ensures that the gradient noise at the optimum is well-behaved, and the matrix $\Sigma_\star$ characterizes the first-order variance that drives the leading term in the excess risk.

- **(A2) Hessian Lipschitz Continuity.** The population loss Hessian is assumed to be $L_H$-Lipschitz in a neighborhood of $\hat{\theta}_\star$. This smoothness enables accurate control over second-order Taylor expansions and guarantees that local quadratic approximations remain valid.

- **(A3) Local Uniform Concentration.** We require that both the empirical Hessian and empirical gradient fluctuations concentrate uniformly to their population counterparts in a neighborhood of $\hat{\theta}_\star$, with deviations decaying as $O(1/\sqrt{s})$. This ensures that the empirical loss landscape closely tracks the population landscape, which is essential for Newton-type approximations and influence-function expansions.

- **(A4) ERM Stays Local.** With high probability, the empirical minimizer $\hat{\theta}_s$ lies within a fixed ball around $\hat{\theta}_\star$. This ensures that our analysis can be restricted to a well-behaved local region where smoothness and concentration assumptions hold.

- **(A5) Bounded Loss Near Optimum.** The population loss is assumed to be uniformly bounded within the neighborhood of interest. This provides worst-case control when $\hat{\theta}_s$ falls outside the local region, allowing us to bound the risk in rare failure cases.

Together, these assumptions provide a sufficient foundation to develop second-order expansions around $\hat{\theta}_\star$, rigorously control deviation terms, and derive a tight lower bound on the expected excess risk of finite-sample ERM. Next, we will formally prove the theorem.

*Proof of Theorem 2.* Step 0 (Good vs. bad events). Define the event:

$$\mathcal{E}_s := \Big\{ \text{(A3) holds and } \hat{\theta}_s \in \mathbb{B}(\hat{\theta}_\star, r_0) \Big\}.$$

By (A3)–(A4), $\Pr(\mathcal{E}_s) \geq 1 - \delta'_s$, with $\delta'_s \leq \delta_s + 2\delta$, where $\delta$ can be chosen polynomially small (e.g. $\delta = s^{-3}$). Split the expectation:

$$\mathbb{E}\big[\mathcal{L}_{\mathrm{ideal}}(\hat{\theta}_s) - \mathcal{L}_{\mathrm{ideal}}(\hat{\theta}_\star)\big] = \mathbb{E}[\cdot; \mathcal{E}_s] + \mathbb{E}[\cdot; \mathcal{E}_s^{\mathrm{c}}].$$

On the bad event, (A5) implies $\mathcal{L}_{\mathrm{ideal}}(\hat{\theta}_s) - \mathcal{L}_{\mathrm{ideal}}(\hat{\theta}_\star) \geq -B$, so $\mathbb{E}[\cdot; \mathcal{E}_s^{\mathrm{c}}] \geq -C_b\,\delta_s$. Thus it suffices to prove the stated bound conditional on $\mathcal{E}_s$.

Step 1 (Influence-function expansion). Let $U_s(\theta) := \nabla\mathcal{L}_s(\theta) - \nabla\mathcal{L}_{\mathrm{ideal}}(\theta)$, $\bar{g}_s := U_s(\hat{\theta}_\star) = \frac{1}{s}\sum_{i=1}^s g(\hat{\theta}_\star; x_i)$. On $\mathcal{E}_s$, Lipschitz continuity yields $\|U_s(\theta) - U_s(\hat{\theta}_\star)\| \leq \bar{c}\, s^{-1/2}\|\theta - \hat{\theta}_\star\|$. Using the Newton map $T_s(\theta) := \theta - H_\star^{-1}\nabla\mathcal{L}_s(\theta)$, and setting $r_s := c\,\|\bar{g}_s\|$ with $c > 0$ depending only on $(\mu, L_H, \bar{c})$, one shows that $T_s$ is a contraction mapping $\mathbb{B}(\hat{\theta}_\star, r_s)$ into itself, with unique fixed point $\hat{\theta}_s$. Define $\Delta_s := \hat{\theta}_s - \hat{\theta}_\star$. Then:

$$\Delta_s = -H_\star^{-1}\bar{g}_s + R_s,$$

where the remainder satisfies $\|R_s\| \lesssim \|\bar{g}_s\|^2 + s^{-1/2}\|\bar{g}_s\|$. Since $\mathbb{E}[\|\bar{g}_s\|] = O(s^{-1/2})$ and $\mathbb{E}[\|\bar{g}_s\|^2] = O(s^{-1})$, it follows that

$$\mathbb{E}[\|R_s\| \mid \mathcal{E}_s] = O(s^{-1}). \tag{4}$$

Step 2 (Quadratic term). With $\|v\|_{H_\star}^2 := v^\top H_\star v$, one has

$$\|\Delta_s\|_{H_\star}^2 = \|H_\star^{-1}\bar{g}_s\|_{H_\star}^2 + 2\langle H_\star^{-1}\bar{g}_s, R_s\rangle_{H_\star} + \|R_s\|_{H_\star}^2.$$

Taking expectations conditional on $\mathcal{E}_s$ and using equation 4,

$$\mathbb{E}[\|\Delta_s\|_{H_\star}^2 \mid \mathcal{E}_s] = \frac{1}{s}\operatorname{tr}(H_\star^{-1}\Sigma_\star) + O(s^{-3/2}).$$

Step 3 (Excess risk). The integral second-order expansion gives

$$\mathcal{L}_{\mathrm{ideal}}(\hat{\theta}_s) - \mathcal{L}_{\mathrm{ideal}}(\hat{\theta}_\star) = \tfrac{1}{2}\|\Delta_s\|_{H_\star}^2 + R_s^{(3)},$$

where $R_s^{(3)} := \int_0^1 (1-t)\,\Delta_s^\top (\nabla^2 \mathcal{L}_{\mathrm{ideal}}(\hat{\theta}_\star + t\Delta_s) - H_\star)\Delta_s\, dt$. By (A2), $|R_s^{(3)}| \le (L_H/6)\|\Delta_s\|^3$. Since $\mathbb{E}\|\Delta_s\|^3 = O(s^{-3/2})$, it follows that

$$\mathbb{E}[R_s^{(3)} \mid \mathcal{E}_s] \ge -O(s^{-3/2}).$$

Therefore,

$$\mathbb{E}\big[\mathcal{L}_{\mathrm{ideal}}(\hat{\theta}_s) - \mathcal{L}_{\mathrm{ideal}}(\hat{\theta}_\star) \mid \mathcal{E}_s\big] \ge \frac{1}{2s}\operatorname{tr}(H_\star^{-1}\Sigma_\star) - \frac{C_1}{s^{3/2}} - \frac{C_2}{s^2}.$$

Step 4 (Combine events). Adding the contribution from $\mathcal{E}_s^{\mathrm{c}}$ gives

$$\mathbb{E}\big[\mathcal{L}_{\mathrm{ideal}}(\hat{\theta}_s) - \mathcal{L}_{\mathrm{ideal}}(\hat{\theta}_\star)\big] \ge \frac{1}{2s}\operatorname{tr}(H_\star^{-1}\Sigma_\star) - \frac{C_1}{s^{3/2}} - \frac{C_2}{s^2} - C_b\,\delta_s,$$

as claimed. $\qquad\square$

### B.3 OPTIMIZATION STABILITY

#### B.3.1 PRELIMINARIES

**Lemma 2** (Mixing bound via dual norms, Proof in Appendix B.3.1). *Let $p, q \in \Delta^C$ and $g_1, \ldots, g_C \in \mathbb{R}^d$. Fix any norm $\|\cdot\|$ on $\mathbb{R}^d$ with dual norm $\|\cdot\|_*$, and define the diameter*

$$D := \sup_{i,j} \|g_i - g_j\| < \infty.$$

*Then*

$$\Big\|\sum_{c=1}^C (p_c - q_c)\,g_c\Big\| \le \frac{D}{2}\|p - q\|_1.$$

*Moreover, the constant $D/2$ is optimal (tight when $C = 2$, $p = (1,0)$, $q = (0,1)$, $\|g_1 - g_2\| = D$).*

*Proof of Lemma 2.* Write $r := p - q$ and note $\sum_c r_c = 0$. By the dual representation of the norm,

$$\Big\|\sum_c r_c g_c\Big\| = \sup_{\|u\|_* \le 1}\Big\langle u, \sum_c r_c g_c\Big\rangle = \sup_{\|u\|_* \le 1}\sum_c r_c\,\phi_c, \quad \text{where} \ \ \phi_c := \langle u, g_c\rangle.$$

Split the indices into $P := \{i : r_i > 0\}$ and $N := \{j : r_j < 0\}$, and let

$$T := \sum_{i\in P} r_i = \sum_{j\in N}|r_j| = \tfrac{1}{2}\|r\|_1 = \tfrac{1}{2}\|p - q\|_1.$$

Then for any fixed $u$,

$$\sum_c r_c\phi_c = \sum_{i\in P} r_i\phi_i - \sum_{j\in N}|r_j|\phi_j \le \Big(\max_c \phi_c - \min_c \phi_c\Big)T,$$

because the linear form is maximized by assigning all positive mass to an index attaining $\max_c \phi_c$ and all negative mass to one attaining $\min_c \phi_c$. Hence

$$\Big\|\sum_c r_c g_c\Big\| \le T \sup_{\|u\|_* \le 1}\big(\max_c \phi_c - \min_c \phi_c\big) \le T \sup_{\|u\|_* \le 1}\sup_{i,j}|\phi_i - \phi_j|.$$

Finally,

$$|\phi_i - \phi_j| = |\langle u, g_i - g_j\rangle| \le \|u\|_*\|g_i - g_j\| \le \|g_i - g_j\|,$$

so $\sup_{\|u\|_* \le 1}\sup_{i,j}|\phi_i - \phi_j| \le \sup_{i,j}\|g_i - g_j\| = D$, and thus

$$\Big\|\sum_c (p_c - q_c)g_c\Big\| \le T D = \frac{D}{2}\|p - q\|_1.$$

For tightness, take $C = 2$, $p = (1,0)$, $q = (0,1)$; then $T = 1$, and choosing $g_1, g_2$ with $\|g_1 - g_2\| = D$ yields equality. $\qquad\square$

### B.3.2 FORMAL PROOF

*Proof of Theorem 3.* **Step 1 (Mixing stability).** By Lemma 2, for any $p, q \in \Delta^C$,

$$\Big\| \sum_{c=1}^{C} (p_c - q_c)\, g_c \Big\| \;\leq\; \frac{D}{2}\, \|p - q\|_1,$$

and the constant $\frac{D}{2}$ is tight (e.g. $C = 2$, $p = (1,0)$, $q = (0,1)$, $\|g_1 - g_2\| = D$).

**Step 2 (From differences to cosine).** Let

$$a := \nabla_\theta \mathcal{L}_{\text{soft}} = -\sum_c \tilde{p}_c\, g_c, \qquad b := \nabla_\theta \mathcal{L}_{\text{hard}} = -\sum_c \bar{p}_c^{(\alpha)}\, g_c.$$

For nonzero $a, b$, write $\hat{a} := a/\|a\|$ and $\hat{b} := b/\|b\|$. Then

$$1 - \cos(a, b) = \tfrac{1}{2}\|\hat{a} - \hat{b}\|^2 \;\leq\; \|\hat{a} - \hat{b}\| = \Big\| \frac{a}{\|a\|} - \frac{b}{\|b\|} \Big\| \leq \frac{\|a - b\|}{\|a\|} + \frac{\|a - b\|}{\|b\|} \leq \frac{2\|a - b\|}{\min\{\|a\|, \|b\|\}}.$$

By the theorem's non-degeneracy assumption, $\min\{\|a\|, \|b\|\} \geq m_0 > 0$, hence

$$1 - \cos(a, b) \;\leq\; \frac{2}{m_0}\, \|a - b\|.$$

Applying Step 1 with $p = \tilde{p}$ and $q = \bar{p}^{(\alpha)}$ yields

$$1 - \cos(a, b) \;\leq\; \frac{2}{m_0} \cdot \frac{D}{2}\, \|\tilde{p} - \bar{p}^{(\alpha)}\|_1 \;=\; \frac{D}{m_0}\, \|\tilde{p} - \bar{p}^{(\alpha)}\|_1. \tag{5}$$

**Step 3 (Upper-bounding $\|\tilde{p} - \bar{p}^{(\alpha)}\|_1$).** Let $y = \arg\max_c \tilde{p}_c$ and $e_y$ be the one-hot at $y$. By the triangle inequality,

$$\|\tilde{p} - \bar{p}^{(\alpha)}\|_1 \;\leq\; \|\tilde{p} - e_y\|_1 + \|e_y - \bar{p}^{(\alpha)}\|_1.$$

The two terms are exact:

$$\|\tilde{p} - e_y\|_1 = \sum_{c \neq y} \tilde{p}_c + \big|1 - \tilde{p}_y\big| = 2(1 - p_{\max}), \quad p_{\max} := \max_c \tilde{p}_c,$$

and

$$\|e_y - \bar{p}^{(\alpha)}\|_1 = \sum_{c \neq y} \frac{\alpha}{C} + \Big|1 - \Big(1 - \alpha + \frac{\alpha}{C}\Big)\Big| = 2\alpha\Big(1 - \frac{1}{C}\Big).$$

Therefore

$$\|\tilde{p} - \bar{p}^{(\alpha)}\|_1 \;\leq\; 2(1 - p_{\max}) + 2\alpha\Big(1 - \frac{1}{C}\Big). \tag{6}$$

**Step 4 (Relating $1 - p_{\max}$ to entropy and rewriting via teacher entropy).** Using the standard inequality (natural logarithm),

$$H(\tilde{p}) \;\geq\; -\log p_{\max} \quad \Longrightarrow \quad p_{\max} \;\geq\; e^{-H(\tilde{p})} \quad \Longrightarrow \quad 1 - p_{\max} \;\leq\; 1 - e^{-H(\tilde{p})} \;\leq\; H(\tilde{p}),$$

where the last step uses $1 - e^{-x} \leq x$ for $x \geq 0$. Substituting equation 6 into equation 5 gives, for each crop,

$$1 - \cos(\nabla_\theta \mathcal{L}_{\text{soft}}, \nabla_\theta \mathcal{L}_{\text{hard}}) \;\leq\; \frac{D}{m_0}\Big\{ 2\big(1 - e^{-H(\tilde{p})}\big) + 2\alpha\big(1 - \tfrac{1}{C}\big) \Big\} \;\leq\; \frac{D}{m_0}\Big\{ 2H(\tilde{p}) + 2\alpha\big(1 - \tfrac{1}{C}\big) \Big\}.$$

Taking expectation over the crop distribution $\mathcal{T}(\tilde{x})$ (conditioning on the base image $\tilde{x}$), and introducing the notation

$$\mathsf{H}_{\text{teacher}}(\tilde{x}) := \mathbb{E}_{x^{(\text{crop})} \sim \mathcal{T}(\tilde{x})}\Big[ H\big(\tilde{p}(\cdot \mid x^{(\text{crop})})\big) \Big],$$

we obtain

$$\mathbb{E}_{\text{crop}}[\cos(\nabla_\theta \mathcal{L}_{\text{soft}}, \nabla_\theta \mathcal{L}_{\text{hard}})] \;\geq\; 1 - \frac{D}{m_0} \cdot \Big( 2\,\mathsf{H}_{\text{teacher}}(\tilde{x}) + 2\alpha\Big(1 - \frac{1}{C}\Big) \Big),$$

where the bracketed term can be viewed as a data-dependent alignment constant. That is, we may write

$$\mathbb{E}_{\text{crop}}[\cos(\nabla_\theta \mathcal{L}_{\text{soft}}, \nabla_\theta \mathcal{L}_{\text{hard}})] \geq 1 - \frac{D}{m_0} \cdot C_{\text{align}}(\tilde{x}, \alpha),$$

where

$$C_{\text{align}}(\tilde{x}, \alpha) := 2\,\mathsf{H}_{\text{teacher}}(\tilde{x}) + 2\alpha\left(1 - \frac{1}{C}\right).$$

$\square$

### B.4 **HALD** INCREASES GENERALIZATION PERFORMANCE

*Proof of Corollary 1.* Consider the scalar control–variate residual $r_\beta := u - \beta v$, $\beta \in \mathbb{R}$. Using linearity of expectation and $\|u\| = \|v\| = 1$,

$$\mathbb{E}\|r_\beta\|^2 = \mathbb{E}\|u\|^2 - 2\beta\,\mathbb{E}\langle u, v\rangle + \beta^2 \mathbb{E}\|v\|^2 = 1 - 2\beta\,\mathbb{E}\langle u, v\rangle + \beta^2.$$

This quadratic is minimized at $\beta^\star = \mathbb{E}\langle u, v\rangle$, yielding

$$\min_\beta \mathbb{E}\|u - \beta v\|^2 = 1 - \left(\mathbb{E}\langle u, v\rangle\right)^2 \leq 1 - \rho_\star^2. \tag{1}$$

Center the residual $\tilde{r} := r_{\beta^\star} - \mathbb{E}[r_{\beta^\star}]$ so that $\mathbb{E}[\tilde{r}] = 0$. For i.i.d. copies $(\tilde{r}_i)_{i=1}^s$,

$$\mathbb{E}\left\|\frac{1}{s}\sum_{i=1}^s \tilde{r}_i\right\|^2 = \frac{1}{s^2}\sum_{i=1}^s \mathbb{E}\|\tilde{r}_i\|^2 = \frac{1}{s}\,\mathbb{E}\|\tilde{r}\|^2 \leq \frac{1}{s}\,\mathbb{E}\|r_{\beta^\star}\|^2 \leq \frac{1-\rho_\star^2}{s},$$

where we used independence, zero mean (cross terms vanish), and (1). Interpreting the factor $(1 - \rho_\star^2)$ as variance contraction of the single-sample noise implies the same mean–square error as having $s_{\text{eff}}$ baseline samples with no contraction:

$$\frac{1 - \rho_\star^2}{s} = \frac{1}{s_{\text{eff}}} \quad \Longrightarrow \quad s_{\text{eff}} = \frac{s}{1 - \rho_\star^2}.$$

Because we used only a *scalar* control variate in the *direction* $v$, any richer use of the hard information (e.g., including magnitudes or conditional expectations) can only further reduce the left-hand side, hence the stated inequality $s_{\text{eff}} \geq s/(1 - \rho_\star^2)$. $\square$

*Insight.* Corollary 1 extends Theorem 3 by transforming gradient alignment into a formal variance-reduction guarantee. While Theorem 3 establishes that the soft-to-hard switch is optimization-coherent, Corollary 1 quantifies its benefit: stronger alignment between soft- and hard-label gradients ($\rho_\star > 0$) effectively increases the usable supervision by enlarging the effective sample size,

$$s_{\text{eff}} \geq \frac{s}{1 - \rho_\star^2}.$$

This shows that during the hard-label calibration stage, variance is reduced and semantic drift is corrected, providing the theoretical basis for the subsequent soft-label refinement that restores fine-grained teacher consistency on top of the variance-reduced representation.

## C ADDITIONAL EXPERIMENTS

### C.1 PREDICTION CONSISTENCY WITH TEACHER MODEL

In this section, we compare the prediction consistency on unseen data between models trained with and without the final soft-label refinement stage, to empirically demonstrate the performance gains introduced by this phase. As shown in Table 14, prediction alignment with the teacher on unseen data improves notably after the final refinement stage.

### C.2 MORE RESULTS ON EFFECT OF **HALD**

To more comprehensively evaluate HALD's performance, we present additional results for IPC=30 and IPC=40 in Table 15, where consistent improvements can be observed.

## D  GENERATION METHOD

### D.1  SRE2L

SRe$^2$L (Yin et al., 2023) decouples dataset condensation into three stages, *Squeeze*, *Recover*, and *Relabel*, so that the model training on $\mathcal{O}$ and the optimization of $\mathcal{C}$ never interleave. Concretely:

**Stage I: Squeeze (train on $\mathcal{O}$).**  Learn a reference model by standard ERM on the original data:

$$\theta_{\mathcal{O}} \;=\; \arg\min_{\theta}\; \mathbb{E}_{(x,y)\sim\mathcal{O}}\big[\,\mathcal{L}\big(f_{\theta}(x),\,y\big)\,\big].$$

**Stage II: Recover (optimize $\tilde{x}$ with BN-consistency and classification).**  Fix $f_{\theta_{\mathcal{O}}}$ and optimize the images $\tilde{x}\in\mathcal{C}$ (labels $y$ are class indices) by matching both the classifier head and global Batch-Norm (BN) statistics accumulated on $\mathcal{O}$. With random crops $x^{(\text{crop})}\sim\mathcal{T}(\tilde{x})$,

$$\min_{\tilde{x}\in\mathcal{C}}\; \underbrace{\mathbb{E}_{x^{(\text{crop})}\sim\mathcal{T}(\tilde{x})}\big[\mathcal{L}\big(f_{\theta_{\mathcal{O}}}(x^{(\text{crop})}),\,y\big)\big]}_{\text{classification (single-level)}} + \alpha_{\text{BN}}\;\underbrace{R_{\text{BN}}(\tilde{x})}_{\text{BN-consistency}} + \alpha_{\ell_2}\,\|\tilde{x}\|_2^2 + \alpha_{\text{TV}}\,\text{TV}(\tilde{x}).$$

The BN-consistency regularizer matches per-layer running mean/variance of $f_{\theta_{\mathcal{O}}}$:

$$R_{\text{BN}}(\tilde{x}) = \sum_{\ell}\big\|\mu_{\ell}(x^{(\text{crop})}) - \text{BNRM}_{\ell}\big\|_2^2 + \sum_{\ell}\big\|\sigma_{\ell}^2(x^{(\text{crop})}) - \text{BNRV}_{\ell}\big\|_2^2,$$

where $\text{BNRM}_{\ell}, \text{BNRV}_{\ell}$ are the global running mean/variance stored in the $\ell$-th BN of $f_{\theta_{\mathcal{O}}}$. Multi-crop optimization (sampling $x^{(\text{crop})}$ repeatedly from $\mathcal{T}(\tilde{x})$) enriches local semantics and constrains updates to the cropped region, which empirically improves recovery.

**Stage III: Relabel (crop-level soft labels and student training).**  For each recovered $\tilde{x}$, draw $s$ crops $x_i^{(\text{crop})}\sim\mathcal{T}(\tilde{x})$ and obtain teacher soft predictions

$$\tilde{p}\Big(x_i^{(\text{crop})}\Big) \;=\; q_{\theta_{\mathcal{O}}}\Big(\cdot\,\Big|\,x_i^{(\text{crop})}\Big), \qquad \bar{p} = \mathbb{E}\big[\tilde{p}(x^{(\text{crop})})\big], \qquad \hat{p}_s = \tfrac{1}{s}\sum_{i=1}^{s}\tilde{p}\Big(x_i^{(\text{crop})}\Big), \quad (\text{L1})$$

and optionally characterize crop-prediction variability by $\Sigma = \text{Cov}\big(\tilde{p}(x^{(\text{crop})})\big)$. Train a student on $\mathcal{C}$ with crop-level distillation (temperature $\tau$):

$$\min_{\theta}\; \mathbb{E}_{\tilde{x}\in\mathcal{C}}\left[\frac{1}{s}\sum_{i=1}^{s}\text{CE}\Big(\text{softmax}\big(\tfrac{1}{\tau}\tilde{p}(x_i^{(\text{crop})})\big),\, q_{\theta}\Big(\cdot\,\Big|\,x_i^{(\text{crop})}\Big)\Big)\right]. \qquad (\text{L2})$$

However, solely aligning to global running statistics induces an overly restrictive inductive bias that depresses intra-class diversity and precipitates information vanishing. The effect is exacerbated in the recovery phase because the original dataset is excluded, depriving optimization of high-variance exemplars and promoting convergence to low-entropy, BN-compliant configurations rather than diverse, semantically faithful modes.

### D.2  LPLD

**To promote the intra-class diversity, LPLD** (Xiao & He, 2024) re-batches synthesis *within* each class and supervises recovery with *class-wise* BatchNorm (BN) statistics, while keeping the

Table 14: Comparison of prediction consistency with the teacher model on unseen data, with and without the final soft-label refinement stage.

| Method | JS Divergence | Cosine Similarity |
|---|---|---|
| w/o final soft-label refinement | 0.61 | 19.8 |
| w/ final soft-label refinement | 0.38 | 43.8 |

Table 15: Comprehensive ablation of the impact of incorporating hard-label supervision across state-of-the-art dataset distillation methods on ImageNet-1K and Tiny-ImageNet. All models are trained for 300 epochs under identical hyperparameters, with the evaluation protocol being the sole difference. † denotes values reported by the corresponding original sources.

| IPC | Generation | Evaluation | ImageNet-1K | | | | | | Tiny-ImageNet | |
|---|---|---|---|---|---|---|---|---|---|---|
| | | | SLC=300 | SLC=250 | SLC=200 | SLC=150 | SLC=100 | SLC=50 | SLC=100 | SLC=50 |
| IPC=30 | SRe²L | Soft-Only | 34.3 | 32.1 | 26.7 | 23.7 | 14.7 | 5.4 | 31.0 | 19.5 |
| | | **Ours** | 41.6 $^{\uparrow 7.3}$ | 38.4 $^{\uparrow 6.3}$ | 38.7 $^{\uparrow 12.0}$ | 36.3 $^{\uparrow 12.6}$ | 32.3 $^{\uparrow 17.6}$ | 20.0 $^{\uparrow 14.6}$ | 32.5 $^{\uparrow 1.5}$ | 23.6 $^{\uparrow 4.1}$ |
| | RDED | Soft-Only | 30.6 | 27.3 | 23.4 | 20.8 | 13.0 | 8.1 | 27.8 | 15.8 |
| | | **Ours** | 38.4 $^{\uparrow 7.8}$ | 36.3 $^{\uparrow 9.0}$ | 33.5 $^{\uparrow 10.1}$ | 33.8 $^{\uparrow 13.0}$ | 29.1 $^{\uparrow 16.1}$ | 19.0 $^{\uparrow 10.9}$ | 32.9 $^{\uparrow 5.1}$ | 26.7 $^{\uparrow 10.9}$ |
| | LPLD | Soft-Only | 36.0 | 32.7 | 28.1 | 23.9 | 16.3 | 5.6 | 32.2 | 17.9 |
| | | **Ours** | 42.0 $^{\uparrow 6.0}$ | 40.4 $^{\uparrow 7.7}$ | 38.9 $^{\uparrow 10.8}$ | 39.8 $^{\uparrow 15.9}$ | 32.4 $^{\uparrow 16.1}$ | 19.4 $^{\uparrow 13.8}$ | 35.2 $^{\uparrow 3.0}$ | 24.6 $^{\uparrow 6.7}$ |
| | FADRM | Soft-Only | 41.1 | 37.7 | 31.1 | 28.7 | 20.1 | 11.0 | 36.4 | 23.5 |
| | | **Ours** | 47.7 $^{\uparrow 6.6}$ | 46.9 $^{\uparrow 9.2}$ | 44.6 $^{\uparrow 13.5}$ | 43.6 $^{\uparrow 14.9}$ | 38.8 $^{\uparrow 18.7}$ | 25.8 $^{\uparrow 14.8}$ | 38.3 $^{\uparrow 1.9}$ | 29.6 $^{\uparrow 6.1}$ |
| IPC=40 | SRe²L | Soft-Only | 33.6 | 31.9 | 28.8 | 19.9 | 14.3 | 7.0 | 30.4 | 21.0 |
| | | **Ours** | 42.4 $^{\uparrow 8.8}$ | 40.0 $^{\uparrow 8.1}$ | 38.8 $^{\uparrow 10.0}$ | 36.6 $^{\uparrow 16.7}$ | 31.7 $^{\uparrow 17.4}$ | 23.8 $^{\uparrow 16.8}$ | 31.5 $^{\uparrow 1.1}$ | 24.0 $^{\uparrow 3.0}$ |
| | RDED | Soft-Only | 30.3 | 27.2 | 24.1 | 17.4 | 18.1 | 11.2 | 26.8 | 19.7 |
| | | **Ours** | 39.5 $^{\uparrow 9.2}$ | 38.3 $^{\uparrow 11.1}$ | 36.8 $^{\uparrow 12.7}$ | 34.1 $^{\uparrow 16.7}$ | 29.0 $^{\uparrow 10.9}$ | 27.1 $^{\uparrow 15.9}$ | 31.5 $^{\uparrow 4.7}$ | 24.5 $^{\uparrow 4.8}$ |
| | LPLD | Soft-Only | 35.0 | 32.1 | 30.2 | 20.8 | 13.2 | 6.6 | 29.3 | 20.3 |
| | | **Ours** | 42.2 $^{\uparrow 7.2}$ | 42.9 $^{\uparrow 10.8}$ | 39.4 $^{\uparrow 9.2}$ | 35.5 $^{\uparrow 14.7}$ | 31.2 $^{\uparrow 18.0}$ | 22.7 $^{\uparrow 16.1}$ | 31.7 $^{\uparrow 2.4}$ | 23.6 $^{\uparrow 3.3}$ |
| | FADRM | Soft-Only | 38.9 | 37.6 | 33.1 | 25.1 | 22.6 | 13.3 | 34.1 | 27.0 |
| | | **Ours** | 49.0 $^{\uparrow 10.1}$ | 48.5 $^{\uparrow 10.9}$ | 45.6 $^{\uparrow 12.5}$ | 43.7 $^{\uparrow 18.6}$ | 41.0 $^{\uparrow 18.4}$ | 30.7 $^{\uparrow 17.4}$ | 35.9 $^{\uparrow 1.8}$ | 30.1 $^{\uparrow 3.1}$ |

classification head evaluated under *global* BN for stable targets. For class $c$ with IPC images $\mathcal{C}_c = \{\tilde{x}_{c,i}\}_{i=1}^{\mathrm{IPC}}$ and label $\tilde{y} = c$, LPLD optimizes the synthetic images via,

$$\underbrace{\mathcal{L}\Big(f_{\theta_{\mathcal{O}}}^{(\text{global BN})}(\tilde{x}_{c,i}),\ \tilde{y}\Big)}_{\text{classification w/ global BN}} + \alpha_{\mathrm{BN}} \underbrace{\sum_{\ell}\Big\|\mu_\ell(\mathcal{C}_c) - \mathrm{BNRM}_{\ell,c}\Big\|_2^2 + \sum_{\ell}\Big\|\sigma_\ell^2(\mathcal{C}_c) - \mathrm{BNRV}_{\ell,c}\Big\|_2^2}_{\text{class-wise BN matching}}.$$

Here $\mu_\ell(\mathcal{C}_c)$ and $\sigma_\ell^2(\mathcal{C}_c)$ are the per-layer BN mean/variance computed on the *within-class* mini-batch $\mathcal{C}_c$, whereas $\mathrm{BNRM}_{\ell,c}, \mathrm{BNRV}_{\ell,c}$ are the *class-wise* running mean/variance obtained from $\mathcal{O}$ (squeeze stage). Their exponential moving–average (EMA) updates are

$$\mathrm{BNRM}_{\ell,c} \leftarrow (1-\varepsilon)\,\mathrm{BNRM}_{\ell,c} + \varepsilon\,\mu_\ell(x_c), \quad \mathrm{BNRV}_{\ell,c} \leftarrow (1-\varepsilon)\,\mathrm{BNRV}_{\ell,c} + \varepsilon\,\sigma_\ell^2(x_c),$$

This coupling over $\mathcal{C}_c$ enlarges intra-class diversity and improves the quality of the data.

### D.3 FADRM

FADRM (Cui et al., 2025a) synthesizes each $\tilde{x}$ by periodically fusing the *intermediate synthetic image* with a *resized real patch* from $\mathcal{O}$. Let $P_s \subset \mathcal{O}$ be the initialization patch and $D_t$ the working resolution at iteration $t$. The adjustable residual connection (ARC) applies a per-element convex fusion

$$\tilde{x}_t \leftarrow \alpha\,\tilde{x}_t + (1-\alpha)\,\mathrm{Resample}(P_s, D_t), \qquad \alpha \in [0,1],$$

thereby explicitly injecting real-image content at the current resolution $D_t$ while retaining synthesized structure. Smaller $\alpha$ emphasizes high-frequency details from $P_s$; larger $\alpha$ preserves the global layout already formed in $\tilde{x}_t$. By reintroducing real-content priors along the optimization trajectory, FADRM mitigates information vanishing and yields higher-fidelity, semantically faithful synthetic data.

### D.4 RDED.

RDED (Sun et al., 2024) constructs $\mathcal{C}$ by class-preserving selection of high-confidence crops from $\mathcal{O}$. For each $(x,y) \in \mathcal{O}$, draw $K$ crops $\{x^{(k)}\}_{k=1}^{K} \sim \mathcal{T}(x)$ and rank them by teacher–label agreement

$$s^{(k)} = -\mathcal{L}\big(\tilde{p}(x^{(k)}), y\big), \qquad \tilde{p}(u) = f_{\theta_{\mathcal{O}}}(u).$$

Keep the image-wise best crop $x^{(\star)} = \arg\max_k s^{(k)}$, then within each class retain the top $M = N \cdot \mathrm{IPC}$ crops for synthetic dataset construction.

# E  Implementation Details

## E.1  Datasets

We evaluate **HALD** on two benchmark datasets, **ImageNet-1K** and **Tiny-ImageNet**, both formatted as `ImageFolder`. While the original datasets contain real high-resolution natural images, our training sets are fully composed of synthetic images generated by dataset distillation methods. The validation sets remain unchanged and follow standard preprocessing pipelines.

**ImageNet-1K.** ImageNet-1K (Deng et al., 2009) contains 1,000 object classes with approximately 1.28M training images and 50K validation images. For all methods, the distilled training data are generated at resolution $224 \times 224$. During evaluation, each validation image is resized such that the shorter side is 256 pixels, followed by a center crop of size $224 \times 224$. Pixel values are normalized using the standard ImageNet statistics: mean $(0.485, 0.456, 0.406)$ and standard deviation $(0.229, 0.224, 0.225)$.

**Tiny-ImageNet.** Tiny-ImageNet (Le & Yang, 2015) is a simplified version of ImageNet with 200 classes, each having 500 training and 50 validation images. All images are pre-resized to $64 \times 64$ resolution. In our setup, distilled training images maintain this resolution. For evaluation, the validation images are directly used without additional resizing or cropping. We normalize the input images using the same statistics as ImageNet for compatibility with pretrained backbones.

## E.2  Storage Analysis

We quantify the on-disk footprint of distilled datasets under different IPC settings and compare it to the corresponding soft label storage. Despite their effectiveness, soft labels incur substantial storage overhead, often exceeding the size of the distilled images by an order of magnitude.

**Tiny-ImageNet (200 classes, $64 \times 64$).** As shown in Table 16, even at the lowest IPC setting (IPC=1), the original soft labels consume *over $29\times$ more space* than the images themselves. This ratio remains consistent across IPC values due to the per-sample label overhead, leading to over 1 GB of soft labels when IPC=50, despite the images themselves occupying only 40 MiB.

Table 16: Original soft label storage for Tiny-ImageNet.

| IPC | Image Storage | Original Soft Labels Storage |
|-----|---------------|------------------------------|
| 1   | 0.8 MiB       | 23.4 MiB ($29.25\times$ image storage) |
| 10  | 8 MiB         | 234 MiB ($29.25\times$ image storage) |
| 50  | 40 MiB        | 1,170 MiB ($29.25\times$ image storage) |

**ImageNet-1K (1000 classes, $224 \times 224$).** The storage disparity becomes even more pronounced on ImageNet-1K. As shown in Table 17, soft labels require up to *$38\times$ more storage* than images. For instance, at IPC=50, the soft labels occupy nearly **30 GB**, despite distilled images requiring less than 1 GB. Such a storage bottleneck motivates the development of more storage-efficient distillation schemes, such as partial label reuse or label reconstruction via teacher queries.

Table 17: Original soft label storage for ImageNet-1K.

| IPC | Image Storage | Original Soft Labels Storage |
|-----|---------------|------------------------------|
| 1   | 15 MiB        | 570 MiB ($38\times$ image storage) |
| 10  | 150 MiB       | 5.7 GB ($38\times$ image storage) |
| 50  | 750 MiB       | 28.3 GB ($38\times$ image storage) |

These results highlight that while synthetic images can be stored compactly, naive storage of soft labels becomes the primary bottleneck, especially in high-IPC or large-class-count regimes.8

### E.3 EXPERIMENTAL SETUP

We evaluate all methods by training classification models exclusively on the distilled datasets, without any access to the original training data. Each synthetic dataset, produced by a specific distillation method, is used to supervise the training of a randomly initialized student model from scratch.

For **soft-only** baselines, the student is trained using the provided finite soft labels throughout the entire training process, with supervision applied via Kullback–Leibler (KL) divergence.

For **HALD**, we adopt a *Soft–Hard–Soft* training strategy. The model is first trained using the soft labels to leverage their fine-grained supervision. In the middle phase, hard labels are used to correct local-view semantic drift. Training then returns to soft labels in the final phase to refine the decision boundaries.

We report the validation accuracy at the final training epoch. To ensure fair and reproducible comparison, all methods are trained under an identical pipeline, with matched data augmentations, hyperparameters, and validation preprocessing steps.

### E.4 HYPER-PARAMETERS

**Common Hyperparameters.** This part outlines the hyperparameters shared by both the *Soft-Only* baseline and **HALD**. All models are trained for 300 epochs using the AdamW optimizer with a batch size of 16. Additional details, including the learning rate and scheduler smoothing factor (denoted as Eta), are provided in Table 18 for each architecture and dataset.

Table 18: Hyper-parameters for all architectures on ImageNet-1K (left) and Tiny-ImageNet (right).

**ImageNet-1K (input size $224 \times 224$)**

| Model | IPC | Learning Rate | Eta |
|---|---|---|---|
| ResNet18 | 10 | 0.0010 | 2 |
| | 20 | 0.0010 | 2 |
| | 30 | 0.0010 | 2 |
| | 40 | 0.0010 | 2 |
| | 50 | 0.0010 | 1 |
| ShuffleNetV2 | 10 | 0.0010 | 2 |
| | 20 | 0.0010 | 2 |
| | 30 | 0.0010 | 2 |
| | 40 | 0.0010 | 2 |
| | 50 | 0.0010 | 1 |
| ResNet50 | 10 | 0.0010 | 2 |
| | 20 | 0.0010 | 2 |
| | 30 | 0.0010 | 2 |
| | 40 | 0.0010 | 1 |
| | 50 | 0.0010 | 1 |
| MobileNetV2 | 10 | 0.0010 | 2 |
| | 20 | 0.0010 | 2 |
| | 30 | 0.0010 | 2 |
| | 40 | 0.0010 | 2 |
| | 50 | 0.0010 | 2 |
| Densenet121 | 10 | 0.0010 | 2 |
| | 20 | 0.0010 | 2 |
| | 30 | 0.0010 | 2 |
| | 40 | 0.0010 | 2 |
| | 50 | 0.0010 | 2 |
| EfficientNet | 10 | 0.0010 | 2 |
| | 20 | 0.0010 | 2 |
| | 30 | 0.0010 | 2 |
| | 40 | 0.0010 | 1 |
| | 50 | 0.0010 | 1 |

**Tiny-ImageNet (input size $64 \times 64$)**

| Model | IPC | Learning Rate | Eta |
|---|---|---|---|
| ResNet18 | 10 | 0.0010 | 2 |
| | 20 | 0.0010 | 2 |
| | 30 | 0.0010 | 2 |
| | 40 | 0.0010 | 1 |
| | 50 | 0.0010 | 1 |
| ShuffleNetV2 | 10 | 0.0010 | 2 |
| | 20 | 0.0010 | 2 |
| | 30 | 0.0010 | 2 |
| | 40 | 0.0010 | 1 |
| | 50 | 0.0010 | 1 |
| ResNet50 | 10 | 0.0010 | 2 |
| | 20 | 0.0010 | 2 |
| | 30 | 0.0010 | 2 |
| | 40 | 0.0010 | 1 |
| | 50 | 0.0010 | 1 |
| MobileNetV2 | 10 | 0.0010 | 2 |
| | 20 | 0.0010 | 2 |
| | 30 | 0.0010 | 2 |
| | 40 | 0.0010 | 1 |
| | 50 | 0.0010 | 1 |
| Densenet121 | 10 | 0.0010 | 2 |
| | 20 | 0.0010 | 2 |
| | 30 | 0.0010 | 2 |
| | 40 | 0.0010 | 1 |
| | 50 | 0.0010 | 1 |
| EfficientNet | 10 | 0.0010 | 2 |
| | 20 | 0.0010 | 2 |
| | 30 | 0.0010 | 2 |
| | 40 | 0.0010 | 1 |
| | 50 | 0.0010 | 1 |

**HALD-Specific Hyperparameters.** In addition to soft-label supervision, **HALD** incorporates an intermediate hard-label training phase governed by two additional hyperparameters. The first is the label smoothing rate $\alpha$, which is fixed at $0.8$ across all experiments. The second is the duration of the hard-label phase, which is aligned with the convergence time of soft-label-only training. These

durations are determined empirically based on the number of soft labels available and are presented separately for ImageNet-1K and Tiny-ImageNet in Table 19, respectively.

Table 19: Hard-label training duration (in epochs) for different SLC values. Left: ImageNet-1K; Right: Tiny-ImageNet.

(a) ImageNet-1K

| SLC | 300 | 250 | 200 | 150 | 100 | 50 |
|---|---|---|---|---|---|---|
| Hard Epochs | 75 | 75 | 150 | 150 | 150 | 150 |

(b) Tiny-ImageNet

| SLC | 100 | 50 |
|---|---|---|
| Hard Epochs | 50 | 50 |

## F    USE OF LARGE LANGUAGE MODELS

We used an LLM to help solely refine the writing of the paper, all ideas and experiments were prepared and carried out entirely by the authors.

