# OpenReview forum: "Hard Labels In! Rethinking the Role of Hard Labels in Mitigating Local Semantic Drift"
_ICLR.cc/2026/Conference — Submitted to ICLR 2026_

### Official Review · Reviewer_x3At · 2025-10-31

**Soundness:** 4
**Presentation:** 3
**Contribution:** 4
**Rating:** 8
**Confidence:** 5

**Summary:**

This paper identifies "Local Semantic Drift" (LVSD), a problem in dataset distillation where using few crops (to save storage) causes soft labels to mismatch the image's global semantics. To address this, the authors propose HALD (Hard Label for Alleviating Local Semantic Drift), a "Soft-Hard-Soft" training schedule that uses hard labels as a "content-agnostic anchor" to correct this drift. The paper provides theoretical justification for the problem and the solution and demonstrates strong empirical results, such as an 8.1% improvement over LPLD on ImageNet-1K under a 285M storage budget.

**Strengths:**

1. Clear Problem Identification: The paper clearly identifies, names, and illustrates (Fig. 1) a practical and important problem (LVSD) that arises from the very real constraint of soft-label storage costs.

2. Strong Empirical Results: When properly isolated (in the ablations), the HALD training schedule provides massive, consistent improvements over a Soft-Only baseline. Table 5 shows HALD boosts performance across all tested generation methods (e.g., +10.5% for FADRM on ImageNet-1K, IPC=10, SLC=100).

3. High Practical Value: The method directly addresses a practical bottleneck (storage). The results in Table 4, showing HALD's robustness to soft-label compression, are highly compelling.

4. Coherent Justification: The paper provides a complete narrative: it identifies a problem (LVSD), formalizes it (Thm 1-2), proposes a solution (HALD), and provides both theoretical (Thm 3, Cor 1) and empirical (Fig 3) justification for why the solution works (gradient alignment leads to variance reduction).

**Weaknesses:**

1. Missing Key Baseline: The paper's novelty rests on its "Soft-Hard-Soft" schedule. However, it fails to compare against the most obvious and simpler baseline: a static combined loss (e.g., $\mathcal{L} = \mathcal{L}_{soft} + \lambda \mathcal{L}_{hard}$), which is related to prior work cited by the authors (e.g., GIFT). This makes it difficult to assess if the complex schedule is truly necessary.

2. The paper's true contribution is best isolated in Table 5, which clearly compares Soft-Only vs. HALD on the same generated data. This table is very strong but is presented as an ablation rather than a main result.

**Questions:**

1. The novelty of the "Soft-Hard-Soft" schedule is a key point. Could the authors provide an ablation comparing HALD against a simpler combined loss (e.g., $\mathcal{L} = \mathcal{L}_{soft} + \lambda \mathcal{L}_{hard\_w/\_LS}$)? This would significantly strengthen the justification for the schedule-based design.

2. The final "Soft" refinement stage (Stage C) appears critical, given the poor performance of "Soft-Hard" in Table 7 (14.2% vs 37.0% for Soft-Hard-Soft). Can you provide more intuition for why this stage is so important?

---

> ### Author Response · Authors · 2025-11-22
> **Response to Reviewer x3At [1/2]**
>
> Thank you for your thoughtful and constructive comments and suggestions. We have uploaded a revised version of our paper that incorporates all concrete changes based on your feedback. Below, we provide detailed responses to each of your concerns.
>
> >W1. Missing Key Baseline: The paper's novelty rests on its "Soft-Hard-Soft" schedule. However, it fails to compare against the most obvious and simpler baseline: a static combined loss (e.g., $\mathcal{L} = \mathcal{L}_{\text{soft}} + \lambda \mathcal{L} _ {\text{hard}}$), which is related to prior work cited by the authors (e.g., GIFT). This makes it difficult to assess if the complex schedule is truly necessary.
>
> We sincerely thank the insightful comment. Following the suggestion, we have provided these two additional baselines that incorporate hard-label information into the soft-only setting for a more comprehensive comparison:
>
> 1. **GIFT**: integrates hard-label information directly into the soft labels using the following formulation:
>
>     $$
>     \tilde{y}_j \leftarrow \gamma \cdot \frac{y_j}{\|y_j\|} + (1 - \gamma) \cdot \frac{\tilde{y}_j}{\|\tilde{y}_j\|},
>     $$
>
>     where $\gamma$ controls the relative weighting between the hard and soft signals. Following the original GIFT setting, we adopt $\gamma = 0.1$, and the refined soft labels are then directly used for model training.
>
> 2. **Joint Objective**: updates the model with a combined loss function that equally leverages both supervision sources:
>
>     $$
>     \mathcal{L} = \mathcal{L}_{\text{soft}} + \lambda \mathcal{L} _ {\text{hard}},
>     $$
>
> where $\lambda$ balances the contributions of the soft- and hard-label objectives.
>
> The results are summarized in the table below. All methods are trained under the same storage and training budgets as HALD for fairness. HALD achieves the best overall performance, while GIFT performs similarly to the soft-only baseline. For the Joint Objective, performance decreases as $\lambda$ increases and peaks at $\lambda=0$ (soft-only), suggesting that mixing hard and soft supervision in a single loss causes gradient inconsistency that degrades performance. As shown in Figure 3 (main paper), the cosine similarity between soft and hard gradients is below 0.4, indicating limited alignment. In contrast, HALD’s stage-wise design leverages their alignment sequentially, avoiding this conflict and achieving consistent improvements. These results confirm that HALD’s advantage arises from its stage-wise integration of soft and hard labels under identical resource constraints. We have included this analysis in **Sec 4.2 (Comparison with more baselines)** of the revised version.
>
>
> |                | SLC=300 |SLC=200| SLC=100 |
> |:--------------|:----:|:----:|:----:|
> |       GIFT  |  46.7    |  39.1    | 27.0|
> | Joint Objective $\lambda=1$|    9.5  |   8.1   | 5.9|
> | Joint Objective $\lambda=0.1$|   10.3   |   9.4   |7.0 |
> | Joint Objective $\lambda=0.01$|    20.1  |   17.5   |13.1|
> | Soft Only      |    46.6  |  39.2    | 26.9|
> | HALD           |    **50.7**  |  **47.3**   | **43.5**    |
>
> >W2. The paper's true contribution is best isolated in Table 5, which clearly compares Soft-Only vs. HALD on the same generated data. This table is very strong but is presented as an ablation rather than a main result.
>
> Thank you for the helpful comment. We agree that Table 5 highlights our key contribution. To improve clarity, we have reorganized the results by moving the content of Table 5 into Tables 1 (Main table) and 2 (Main table). To keep the ablation section concise, we have moved the IPC 30 and IPC 40 results to the **Appendix C.2**. This restructuring makes the paper’s contribution clearer and avoids redundancy.

---

> ### Author Response · Authors · 2025-11-22
> **Response to Reviewer x3At [2/2]**
>
> >Q1. The novelty of the "Soft-Hard-Soft" schedule is a key point. Could the authors provide an ablation comparing HALD against a simpler combined loss (e.g., $\mathcal{L} = \mathcal{L}_{\text{soft}} + \lambda \mathcal{L} _ {\text{hard}}$)? This would significantly strengthen the justification for the schedule-based design.
>
> Thanks and please check our response to Weakness 1 for details. We also have integrated them in **Sec 4.2 (Comparison with more baselines)** of our revised paper.
>
> >Q2. The final "Soft" refinement stage (Stage C) appears critical, given the poor performance of "Soft-Hard" in Table 7 (14.2% vs 37.0% for Soft-Hard-Soft). Can you provide more intuition for why this stage is so important?
>
> We thank the reviewer for this thoughtful question. We begin by clarifying the functionality of each stage in *Soft–Hard–Soft*. The initial soft-label stage aligns the student model with the teacher’s semantic distribution and enhances gradient similarity with hard labels, as shown in Theorem 3. The subsequent hard-label calibration stage reduces prediction variance and corrects semantic drift, as formalized in Corollary 1. Finally, the last soft-label refinement stage restores fine-grained teacher consistency on top of the variance-reduced representation. This final phase is crucial: although hard-label calibration effectively reduces variance, it disrupts the alignment with the teacher’s prediction distribution, which can degrade generalization. The last refinement phase therefore reintroduces the teacher's distributional details while maintaining the benefits of variance reduction, striking a balance between reliance on limited soft labels and global semantics from hard labels, thereby improving performance. As shown in the table below, the model after the final refinement stage achieves higher consistency with the teacher’s predictions on unseen data compared to the model trained only with hard-label calibration. We have refined our intuition on why last soft refinement stage is important in **Sec 3.3**, and we have added the experiment in **Appendix C.1**.
>
> |  | JS Divergence | Cosine Similarity|
> |:-------|:---------------:|:------------------:|
> | w/o final soft label refinement| 0.61 | 19.8 |
> | w/ final soft label refinement | **0.38** | **43.8** |

---

### Official Review · Reviewer_xHep · 2025-11-01

**Soundness:** 3
**Presentation:** 3
**Contribution:** 2
**Rating:** 4
**Confidence:** 4

**Summary:**

This paper explores the local semantic drift problem when applying soft labels in dataset distillation methods for downstream training, pointing out that when the number of image croppings is limited, soft labels may deviate from the original semantics, resulting in a mismatch between training and test distributions. To address this issue, this paper proposes HALD method, which employs a three-stage soft-hard-soft training mechanism. First, hard labels are introduced as content-agnostic anchors to calibrate semantic drift, building upon soft label training. Then, the training returns to soft labels for fine-tuning to maintain fine-grained semantic features. Also, the theoretical analysis in the paper explains that introducing hard labels into soft-label training can improve optimization stability and generalization ability.

**Strengths:**

1. The paper points out the problem of local semantic drift in soft-label based dataset distillation with a clear motivation.
2. The paper also theoretically analyzes the importance of introducing hard labels into soft-label based dataset distillation, demonstrating its ability to improve optimization stability and generalization ability. The paper is well-reasoned and supported.
3. The performance of the proposed method compared with baseline under the same storage budget is good.
4. The article has a good structure and is well-written.

**Weaknesses:**

1. When SLC is set to a fixed value (e.g., 300), the performance of the original method (which requires augmentation and generation of corresponding soft labels in each epoch during downstream training) is degrade. However, SRe2L is improved compared to the original method. This phenomenon is strange (because according to Table 5, the performance actually decreases when SLC is reduced).
2. One question is that, according to the original method, methods like RDED can directly store the teacher model for online soft label generation. For example, the ResNet-18 model only requires 44.7MB of storage, while IPC10 can achieve an accuracy of 42%.
3. There are also some methods, e.g., GIFT[1], for combining soft labels and hard labels, but the paper does not make further comparisons, only briefly described in related work part. Other methods [2] explored the domain shift issue, but no further comparisons are made.

[1] GIFT: Unlocking Full Potential of Labels in Distilled Dataset at Near-zero Cost

[2] Large Scale Dataset Distillation with Domain Shift

**Questions:**

1. Based on weaknesses 1 and 2, it would be helpful if the author could provide further explanation and clarification regarding these questions. Furthermore, it would be appreciated if the author could clearly explain how to choose the timing (not the length) for hard-label training.
2. Based on weakness 3, if the authors can provide further comparisons and explanations would be appreciated.
3. The cross-architecture experiments mainly performed on residual-based models. To demonstrate a more comprehensive cross-architecture capability, comparisons can be made with models such as ViT and VGG.

---

> ### Author Response · Authors · 2025-11-22
> **Response to Reviewer xHep [1/2]**
>
> Thank you for your thoughtful and constructive comments and suggestions. We have uploaded a revised version of our paper that incorporates all concrete changes based on your feedback. Below, we provide detailed responses to each of your concerns.
>
>
> >W1. When SLC is set to a fixed value (e.g., 300), the performance of the original method (which requires augmentation and generation of corresponding soft labels in each epoch during downstream training) is degrade. However, SRe2L is improved compared to the original method. This phenomenon is strange (because according to Table 5, the performance actually decreases when SLC is reduced).
>
> We appreciate the reviewer's insightful observation on this result. We clarify that the training configuration used in the original SRe$^2$L [1] paper was suboptimal, which led to degraded performance. Consequently, SRe$^2$L exhibits reduced performance under limited soft-label coverage compared to its reference model counterpart trained with full soft-label coverage. The detailed comparison is provided in the table below.
>
> | full soft label converage |SLC=300 |SLC=200| SLC=100 |
> |:----:|:----:|:----:|:----:|
> |  43.1  |  32.2   | 28.0 | 18.6|
>
> >W2. One question is that, according to the original method, methods like RDED can directly store the teacher model for online soft label generation. For example, the ResNet-18 model only requires 44.7MB of storage, while IPC10 can achieve an accuracy of 42%.
>
> Thanks for raising this question. We clarify that our motivation for storing pre-computed soft labels instead of the teacher model is to preserve the standard dataset format and improve efficiency. First, a standard dataset consists of *image–label* pairs, not *image–teacher* pairs. Involving a teacher during student training alters this paradigm, as a distilled dataset should be self-contained and reusable without external supervision.  Second, storing the teacher model for online soft-label generation introduces additional computational overhead. Each iteration would require forward passes through the teacher network, and thus increasing training time and hardware costs, especially with large teacher backbones, which contradicts the goal of dataset distillation: achieving **high efficiency** in both storage and computation. In contrast, HALD pre-computes and stores soft labels offline. This design enables faster training without relying on the teacher and avoids repeated teacher inference that would offset the benefit of smaller model storage, while also allowing flexible control of storage via the SLC budget.
>
>
> >W3. There are also some methods, e.g., GIFT [1], for combining soft labels and hard labels, but the paper does not make further comparisons, only briefly described in related work part. Other methods like D3S [2] explored the domain shift issue, but no further comparisons are made.
>
> We appreciate the comment to include additional comparisons. Following the suggestion, we incorporate two additional baselines for evaluation: **GIFT** [1] and **Joint Objective**, where the latter updates the model with a combined loss function that equally leverages both supervision sources: $\mathcal{L} = \mathcal{L}_{\text{soft}} + \lambda \mathcal{L} _ {\text{hard}},$ where $\lambda$ balances the contributions of the soft- and hard-label objectives. We kindly note that D3S [2] focuses on generating distilled images rather than on post-evaluation, which makes it not directly comparable to HALD.
>
>
> The experimental results are summarized in the table below. All methods are trained under the same storage and training budget as HALD to ensure fairness. We observe that our **HALD achieves the best overall performance**. In contrast, **GIFT** provides little improvement and performs similarly to the Soft-Only baseline. For the **Joint Objective**, performance decreases as $\lambda$ increases and peaks at $\lambda=0$ (the Soft-Only case). This indicates that directly mixing hard and soft supervision within the same loss introduces gradient inconsistency degrades performance. We have added this analysis in **Sec 4.2 (Comparison with more baselines)** in the revision.
>
>
> |                | SLC=300 |SLC=200| SLC=100 |
> |:--------------|:----:|:----:|:----:|
> |       GIFT [1]  |  46.7    |  39.1    | 27.0|
> | Joint Objective $\lambda=1$|    9.5  |   8.1   | 5.9|
> | Joint Objective $\lambda=0.1$|   10.3   |   9.4   |7.0 |
> | Joint Objective $\lambda=0.01$|    20.1  |   17.5   |13.1|
> | Soft-Only      |    46.6  |  39.2    | 26.9|
> | HALD           |    **50.7**  |  **47.3**   | **43.5**    |

---

> ### Author Response · Authors · 2025-11-22
> **Response to Reviewer xHep [2/2]**
>
> >Q1. Furthermore, it would be appreciated if the author could clearly explain how to choose the timing (not the length) for hard-label training.
>
> Thanks for raising this important point. Our theoretical analysis assumes that the model can converge properly under limited soft labels. Once this assumption holds, Corollary 1 indicates that hard-label semantic calibration increases the effective sample size, thereby reducing variance and mitigating LVSD. Based on this, we define the total duration of the soft-label phases in the *Soft–Hard–Soft* schedule as the training time required for the model to converge under limited soft labels. As shown in the table below, the first and last soft phases are equally important, since allocating more training budget to either phase leads to inferior performance compared with the balanced setting. The first phase ensures convergence and a stable transition by improving gradient alignment, while the last phase restores fine-grained teacher consistency on the variance-reduced representation. Therefore, we equally divide the total soft-label duration between the two soft phases, allocate the remaining time to the hard-label phase, and switch to hard supervision once the first soft phase is done. We have added this experiment in **Sec 4.5 (Effect of the first and last soft-label stages)** and revised how we choose the timing in **Sec 3.3**.
>
> | First Soft Phase duration | Last Soft Phase duration | HALD Performance |
> |:----:|:----:|:----:|
> |  50    |  100    | 35.2 |
> |  100   |   50    | 34.7|
> |  75    |   75    | **35.6** |
>
>
>
> >Q2. Based on weakness 3, if the authors can provide further comparisons and explanations would be appreciated.
>
> We appreciate the reviewer for the kind and thoughtful question. The corresponding results are presented below, and a detailed description of the baseline settings is provided in our response to Weakness 3 and our revised paper.
>
> |                | SLC=300 |SLC=200| SLC=100 |
> |:--------------|:----:|:----:|:----:|
> |       GIFT [1]  |  46.7    |  39.1    | 27.0|
> | Joint Objective $\lambda=1$|    9.5  |   8.1   | 5.9|
> | Joint Objective $\lambda=0.1$|   10.3   |   9.4   |7.0 |
> | Joint Objective $\lambda=0.01$|    20.1  |   17.5   |13.1|
> | Soft-Only      |    46.6  |  39.2    | 26.9|
> | HALD           |    **50.7**  |  **47.3**   | **43.5**    |
>
> >Q3. The cross-architecture experiments mainly performed on residual-based models. To demonstrate a more comprehensive cross-architecture capability, comparisons can be made with models such as ViT and VGG.
>
> Thanks for the suggestion. We have further evaluated HALD on additional backbone architectures, as shown in the table below. Consistent with our previous findings, HALD continues to outperform the soft-only baseline across different architectures. We have added new result in **Table 3**.
>
> | Model | RDED + Soft Only | LPLD + Soft Only | FADRM + Soft Only | **HALD** |
> |:------|:----------------:|:----------------:|:-----------------:|:--------:|
> | **ViT-Tiny** | 3.2 | 5.6 | 5.6 | **8.9** |
> | **VGG-11**   | 26.3 | 28.9 | 31.0 | **33.6** |
> | **VGG-16**   | 28.9 | 34.3 | 36.2 | **37.4** |
>
> -----------
> References:
>
> [1] Yin, Zeyuan, Eric Xing, and Zhiqiang Shen. "Squeeze, recover and relabel: Dataset condensation at imagenet scale from a new perspective." In NeurIPS 2023.
>
> [2] Loo, Noel, Alaa Maalouf, Ramin Hasani, Mathias Lechner, Alexander Amini, and Daniela Rus. "Large scale dataset distillation with domain shift." In ICML 2024.

---

### Official Review · Reviewer_6Y9U · 2025-11-03

**Soundness:** 2
**Presentation:** 3
**Contribution:** 2
**Rating:** 2
**Confidence:** 3

**Summary:**

This paper studies dataset distillation under low Soft Label per Class (SLC) and without further teacher access. It is observed that local crops of an image can obtain soft targets that are inconsistent with the global image label. In this paper, it is referred to as Local View Semantic Drift (LVSD). To address this, the authors propose a three-stage training schedule, named HALD, which consists of Soft, Hard, and Soft stages. On the specific protocol defined in the paper, HALD achieves strong empirical results, as shown in Table 4.

**Strengths:**

- The paper identifies a concrete failure mode of low SLC distillation. Local crops do not always agree with the image-level target;
- The proposed Soft Hard Soft schedule is simple and easy to add to existing dataset distillation pipelines;
- Empirical results in the stated setting are clearly positive. The paper also attempts to give a theoretical view, which is not very common in this area.

**Weaknesses:**

### The research problem depends on a specific setting
The whole story assumes that the image-level hard label is the only correct semantic anchor. Any crop level target that is different from it is treated as drift. This is one possible viewpoint, but not the only reasonable one. If we look from a local visual viewpoint, then many of the paper’s examples are no longer drift. A crop that shows only the ball in a human and ball image can reasonably receive the class ball from the teacher. In this viewpoint HALD is not correcting an error. HALD is enforcing a preference for global semantics over local visual fidelity.

The paper should explain why the global label dominant viewpoint is the right one for dataset distillation. Otherwise, it is more like you first chose a particular way of looking at the problem, and that viewpoint makes the issue look serious. But if someone does not adopt your viewpoint, the issue may not even arise. Therefore, you cannot present it as a universally occurring problem.

### The reason for the performance improvement is unclear
HALD is described as using image-level hard labels as a content-agnostic anchor to pull drifting soft labels back. In many realistic images, the global hard label is not an unbiased anchor for a local crop. It is another biased signal at a different scale. For example, in a multi-object scene or in a heavily occluded crop, the global label can be semantically farther from the crop than the original soft prediction.

A very plausible interpretation is therefore the following. Stage B injects a deliberately mismatched but stable supervision signal, which prevents the student from overfitting any single view and thus acts as a strong regularizer. The current experiments do not distinguish between these two explanations. If the authors want to claim semantic calibration, they need to show crop-level consistency or similar observable quantities before and after Stage B. Otherwise, the safe reading is that HALD is a good regularized schedule rather than a semantic correction mechanism.

### Alignment between theory and Algorithm
Section 3.4 together with Appendix B.4 proves a control variate style variance reduction statement. In that setting, two highly correlated signals are used in the same update. The result is an effective sample size of at least s over $1- \rho^2$. The implemented algorithm is different. HALD uses the two signals in different stages. First soft labels, then hard labels, then soft labels again. The paper does not provide a mathematical argument that this stage-wise or asynchronous use of correlated signals preserves the same effective sample size gain as the synchronous case analyzed in the theory.

Another thing. Theorem 3 on gradient alignment only shows that the switch is optimization coherent. It shows that the gradients of soft and hard become close, so the switch will not destroy training. It does not show that the switch can improve performance or reduce variance in the way claimed for the per-step setting. At present, the theory should be read as motivational for HALD rather than as a formal justification of the exact algorithm in Section 3.3.

### Experiments
All main tables compare HALD, which uses soft and hard signals in stages, with a soft-only baseline under low SLC and without teacher access. This is an information source un-equal comparison. Two supervision sources are compared to one supervision source. A stronger test would keep storage and training budget the same and add a simultaneous objective $L = L_{soft} + \lambda L_{hard}$ or allow existing baselines to attach the same hard label branch. Without this comparison, we cannot tell whether the gain comes from the schedule itself or simply from the fact that HALD uses one more supervision signal.

**Questions:**

- Can you report a simultaneous soft plus hard baseline under the same settings?
- The variance reduction derivation in Appendix B.4 also applies to the stage-wise HALD, or is it only an intuitive guide?
- What happens if the teacher can produce crop-level or region-level predictions so that LVSD is small? If we adopt a relaxed setting where the teacher can be queried a few times during training, is HALD still the best use of the budget?

---

> ### Author Response · Authors · 2025-11-22
> **Response to Reviewer 6Y9U [1/3]**
>
> Thank you for your thoughtful and constructive feedback. We have uploaded a revised version of our paper incorporating all concrete revisions following your suggestions. We would like to further provide detailed point-by-point responses to address your concerns as follows:
>
> > W1. The research problem depends on a specific setting.
>
> We appreciate the reviewer's insightful comment. We would like to clarify that our work does not assume that global semantics are inherently more accurate than local teacher predictions. Instead, we use the model trained with full soft-label coverage as the reference. Our study focuses on the limited soft-label scenario, where Local-View Semantic Drift (LVSD), an intrinsic property of teacher models under strong augmentation, introduces inacurate supervision that degrades student performance. We identify this issue and incorporate global semantics to mitigate LVSD through increased effective sample size, thereby enabling the limited soft labels to provide more accurate supervision.
>
> > W2. The reason for the performance improvement.
>
> Thanks for raising this important point. We confirm that the performance improvement of HALD indeed originates from the semantic calibration phase (Stage B), rather than from the strong regularization effect of a mismatched but stable signal that might prevent the student from overfitting to a single view, as suggested by the reviewer.
>
> To directly test the reviewer's hypothesis, we replaced the ground-truth hard-label supervision in Stage B with randomly assigned labels at the image level, each image was permanently assigned a random label that remained unchanged throughout training (serve as mismatched but stable supervision signal as reviewer mentioned). This setting preserves the stability of supervision while removing semantic alignment, serving as a stability-preserving control. If the observed gain came from strong regularization, performance should remain similar or even improve. However, we observed a substantial performance collapse, clearly indicating that HALD's improvement does not arise from a regularization effect.
>
>
> |IPC| SLC | Random | HALD |
> |:----:|:-------:|:----:|:----:|
> | 10 |100 |    20.8     |   **35.6**   |
> | 10 |200 |     35.6    |   **40.7**   |
> | 30 |100 |    12.5     |    **43.5** |
> | 30 |200 |    40.1     |   **47.3**   |
>
>
> To further validate that HALD's performance gain stems from semantic calibration, we analyze both crop-level consistency and prediction alignment with a reference model trained under full soft-label coverage. Specifically, crop-level consistency quantifies how well predictions from different crops of the same image agree, measured by the average Jensen--Shannon (JS) divergence and cosine similarity before and after Stage B. Prediction alignment, on the other hand, evaluates how closely the student model's predictions (Soft-Only or HALD) match those of the reference model on unseen data. As shown in the table below, we observed improved semantic consistency and stronger prediction alignment with the reference model, validating the role of hard labels in mitigating semantic drift and improving overall performance.
>
> **Average Crop-level consistency**
>
> |         | JS Divergence  |Cosine Similarity|
> |:-------|:----:|:----:|
> |Before Stage B |  0.1811      |     0.744 |
> |After Stage B |   **0.0393**      |   **0.959**   |
>
> **Average Prediction Similarity with Reference Model**
>
> | Method | JS Divergence | Cosine Similarity|
> |:-------|:---------------:|:------------------:|
> | w/o Hard Calibration | 0.337 | 0.458 |
> | w/ Hard Calibration | **0.226** | **0.623** |
>
>
> These additional analyses collectively clarify that the improvement originates from semantic calibration, as both necessary (performance collapse under random labels) and observable (enhanced semantic consistency) conditions are satisfied. We have added this analysis in **Sec 4.3 (Semantic Calibration.)** in the revision.

---

> ### Author Response · Authors · 2025-11-22
> **Response to Reviewer 6Y9U [2/3]**
>
> >W3. Alignment between theory and Algorithm.
>
> Thanks for the suggestion. We clarify that **Theorem 3 analyzes exactly the same setting as HALD**. The theorem does **not** assume simultaneous updates of two correlated signals; rather, it examines how, during training with either type of label, the gradients of soft- and hard-label losses become increasingly aligned. We have revised **Sec 3.4** to improve clarification.
>
> **Corollary 1** then extends Theorem 3 by converting this gradient alignment into a formal variance-reduction guarantee. While Theorem 3 establishes that the soft-to-hard switch is optimization-coherent, Corollary 1 proves that such switch quantitatively increases the effective sample size:
>
> $$
> s_{\text{eff}} \ge \frac{s}{1 - \rho_\star^2},
> $$
>
> where $\rho_\star$ is the expected cosine similarity defined in Theorem 3. The complete *Soft->Hard->Soft* design naturally follows from this theoretical foundation: the initial **soft-label stage** aligns the model with the teacher's semantic distribution and strengthens gradient similarity with hard labels;  the **hard-label calibration stage** reduces variance and corrects semantic drift as formalized in Corollary 1; and the final **soft-label refinement** restores fine-grained teacher consistency on top of the variance-reduced representation. We have refined **Sec. 3.4** to clarify this point more explicitly.
>
>
>
> >W4 & Q1. Experiments & comparison with more baseline methods.
>
> We thank the reviewer for the suggestion. To address this comment, we incorporate two additional baselines that integrate hard-label information into the soft-only setting, following the reviewer's suggestion to ensure a more balanced comparison:
>
> 1. **GIFT [1]**: integrates hard-label information directly into the soft labels using the following formulation:
>
>     $$
>     \tilde{y}_j \leftarrow \gamma \cdot \frac{y_j}{\|y_j\|} + (1 - \gamma) \cdot \frac{\tilde{y}_j}{\|\tilde{y}_j\|},
>     $$
>
>     where $\gamma$ controls the relative weighting between the hard and soft signals. Following the original GIFT setting, we adopt $\gamma = 0.1$, and the refined soft labels are then directly used for model training.
>
>
>
> 2. **Joint Objective**: updates the model with a combined loss function that equally leverages both supervision sources:
>
>     $$
>     \mathcal{L} = \mathcal{L}_{\text{soft}} + \lambda \mathcal{L} _ {\text{hard}},
>     $$
>
>     where $\lambda$ balances the contributions of the soft- and hard-label objectives.
>
> The results are summarized in the table below. All methods are trained under the same storage and training budgets as HALD for fairness. HALD achieves the best overall performance, while GIFT performs similarly to the soft-only baseline. For the Joint Objective, performance decreases as $\lambda$ increases and peaks at $\lambda=0$ (soft-only), suggesting that mixing hard and soft supervision in a single loss causes gradient inconsistency that degrades performance. As shown in Figure 3 (main paper), the cosine similarity between soft and hard gradients is below 0.4, indicating limited alignment. In contrast, HALD's stage-wise design leverages their alignment sequentially, avoiding this conflict and achieving consistent improvements. These results confirm that HALD's advantage arises from its stage-wise integration of soft and hard labels under identical resource constraints. We have included this analysis in **Sec 4.2 (Comparison with more baselines)** of the revised version.
>
>
> |                | SLC=300 |SLC=200| SLC=100 |
> |:--------------|:----:|:----:|:----:|
> |       GIFT [1]  |  46.7    |  39.1    | 27.0|
> | Joint Objective $\lambda=1$|    9.5  |   8.1   | 5.9|
> | Joint Objective $\lambda=0.1$|   10.3   |   9.4   |7.0 |
> | Joint Objective $\lambda=0.01$|    20.1  |   17.5   |13.1|
> | Soft Only      |    46.6  |  39.2    | 26.9|
> | HALD           |    **50.7**  |  **47.3**   | **43.5**    |

---

> ### Author Response · Authors · 2025-11-22
> **Response to Reviewer 6Y9U [3/3]**
>
> >Q2. The variance reduction derivation in Appendix B.4 also applies to the stage-wise HALD, or is it only an intuitive guide?
>
> Thanks for the comment. The variance-reduction derivation in Appendix B.4 **also applies to the stage-wise HALD setting**, as detailed in our response to Weakness 3. We have also added analysis in **Appendix B.4** to better clarify this point.
>
> >Q3 (1). What happens if the teacher can produce crop-level or region-level predictions so that LVSD is small?
>
> Thanks for this insightful question. We conclude that the degree of Local-View Semantic Drift (LVSD) is consistently large across teacher models, caused by the strong augmentation used in training. To quantify this, we define the LVSD ratio:
>
> $$
> R(\tilde{x}) = \frac{\mathrm{Tr}(\hat{\Sigma}_{\text{strong}})}{\mathrm{Tr}(\hat{\Sigma} _ {\text{weak}}) + \varepsilon},
> $$
>
> where the numerator measures prediction variance under strong augmentation (aggressive random resized cropping and flipping), and the denominator under weak augmentation (resize + center crop). Because our setup relies on strong augmentation by default, $R(\tilde{x})$ captures the drift induced by local views. As shown below, $R$ is extremely large across ResNet-18, MobileNetV2, and ShuffleNetV2, confirming that LVSD is substantial and cannot be ignored. We have added these analysis in **Sec 4.3 (LVSD Quantification)** in our revised paper.
>
> |        Teacher        | Mean $\mathrm{Tr}(\hat{\Sigma}_{\text{weak}})$|Mean $\mathrm{Tr}(\hat{\Sigma}_{\text{strong}})$| $\log_{10}(\text{Mean } R)$| $\mathbf{p}(R > 1)$|
> |:--------------|:----:|:----:|:---:|:---:|
> |    ResNet-18  |  4.8356 $\times$ $10^{-15}$   |  0.0102    | 3.27    | 97.2 %|
> |    MobileNet-V2  |    4.4225 $\times$ $10^{-15}$    |  0.3756       |  5.28  | 99.2 % |
> |    ShuffleNet-V2 |   4.5138 $\times$ $10^{-15}$     |   0.0591      |   5.35   | 98.0% |
>
>
>
> >Q3 (2). If we adopt a relaxed setting where the teacher can be queried a few times during training, is HALD still the best use of the budget?
>
> Thanks for the careful observation and interesting question. Querying the teacher model during training essentially increases the effective storage budget for soft labels, i.e., the SLC. Under the limited soft-label setting, a fixed SLC corresponds to a fixed supervision budget. Therefore, when the budget is held constant, applying HALD as the evaluation method consistently yields higher performance than the Soft-Only baseline, as shown in the table below. This indicates that HALD is the most effective use of a fixed supervision budget.
>
> | Generation | Evaluation | SLC=300 | SLC=200 | SLC=100 |
> |:-----------|:-----------:|:-------:|:-------:|:-------:|
> | SRe$^2$L | Soft-Only | 32.2 | 28.0 | 18.6 |
> |SRe$^2$L  | **HALD** | **35.2** | **33.3** | **31.7** |
> | RDED | Soft-Only | 26.2 | 21.4 | 18.1 |
> | RDED | **HALD** | **27.2** | **25.5** | **25.4** |
> | LPLD | Soft-Only | 32.7 | 32.7 | 23.1 |
> | LPLD | **HALD** | **37.0** | **33.9** | **28.8** |
> | FADRM | Soft-Only | 42.1 | 39.0 | 26.5 |
> | FADRM | **HALD** | **43.4** | **40.4** | **37.0** |
>
>
>
>
> ----------------------
> References:
>
> [1] Xinyi Shang, Peng Sun, Tao Lin. GIFT: Unlocking Full Potential of Labels in Distilled Dataset at Near-zero Cost. ICLR 2025.

---

### Author Response · Authors · 2025-12-03
**Summary for Reviews and Rebuttals**

Dear Reviewers and AC,

We would like to express our sincere gratitude for the time, effort, and thoughtful evaluation that you and all reviewers dedicated to our submission. We truly appreciate the constructive feedback and are grateful that multiple strengths of our work were recognized. These include the simplicity and ease of integrating our method into existing dataset distillation pipelines (**Reviewer 6Y9U**), the strong empirical results (**Reviewers 6Y9U, xHep, x3At**), the sound theoretical analysis (**Reviewers xHep, x3At**), the clear motivation (Reviewer xHep), the well-reasoned and well-supported methodology (**Reviewer xHep**), the solid structure and writing quality (**Reviewer xHep**), the clear problem formulation (**Reviewer x3At**), and the high practical value of the work (**Reviewer x3At**).

**Reviewer x3At** provided particularly positive assessments, and we have submitted comprehensive rebuttals and additional experiments addressing all their raised concerns.


**Reviewer 6Y9U** initially assigned a score of 2 and raised five concerns: (1) the problem’s reliance on a specific setting, (2) the source of the performance improvement, (3) the alignment between theory and the algorithm, (4) comparisons with additional baselines, and (5) the quantification of LVSD.

1. For Concern 1, we have clarified that our method does not rely on the assumption that global semantics are inherently more accurate than local teacher predictions.
2. For Concern 2, we have provided additional empirical evidence demonstrating that HALD’s performance improvements are primarily attributable to semantic calibration.
3. For Concern 3, we have elaborated that the theorem does not presume simultaneous updates of two correlated signals, thereby establishing a strong consistency between the theoretical analysis and the algorithmic design.
4. For Concern 4, we have expanded our experimental comparisons by including additional baselines, further substantiating the effectiveness of HALD.
5. For Concern 5, we have introduced quantitative analyses of LVSD and verified that its magnitude is both meaningful and significant.


**Reviewer xHep** initially assigned a score of 4 and raised four concerns: (1) the behavior of SRe$^2$L under limited soft-label availability, (2) the advantages of storing pre-generated soft labels, (3) the need for comparisons with additional baselines, and (4) the request for more extensive cross-architecture evaluations.

1. For Concern 1, we have clarified that SRe²L’s performance gains under small SLC budgets arise from the suboptimal training configuration used in the original paper.
2. For Concern 2, we have articulated the primary advantages of storing pre-generated soft labels over online generation.
3. For Concern 3, we have broadened our experimental comparisons by incorporating additional baselines, thereby further substantiating the effectiveness of HALD.
4. For Concern 4, we have evaluated a wider range of architectures to more comprehensively demonstrate the robustness and effectiveness of HALD.

We sincerely appreciate the reviewers' thoughtful suggestions and feedback during the review process, and we hope that our summary clarifications above and detailed responses below have addressed all reviewers' concerns. We have also uploaded a revision of our manuscript that incorporates all reviewer comments.

Best regards,

Authors of submission 13047

---

### Meta-Review · Area_Chair_bhnv · 2026-01-07

**Summary:**

The paper introduces HALD, a three-stage training schedule designed to address "Local Semantic Drift" in dataset distillation, a phenomenon where local image crops receive mismatched soft labels from a teacher model under limited storage budgets. While the paper identifies a practical bottleneck in offline distillation, the proposed solution is essentially a heuristic training recipe. The reviewers raised concerns regarding the fundamental novelty of using hard labels as anchors, the sensitivity of the three-stage timing, and the narrow applicability of the setting (offline distillation with fixed storage).

**Reviewer Concerns:**

- Heuristic vs. Theoretical Gap: While the authors provide a variance-reduction analysis (Corollary 1), there remains a significant logical leap between the mathematical theory and the specific three-stage schedule. The theory justifies the use of hard labels but does not explain why an asynchronous, stage-wise transition is superior to more principled, unified optimization frameworks.

- Incremental Technical Novelty (Reviewer 6Y9U, xHep): The observation that local crops can deviate from global labels (LVSD) is a well-known characteristic of data augmentation and teacher-student misalignment. Utilizing hard labels to regularize or "anchor" soft labels is a common practice in knowledge distillation. The contribution is largely reduced to a tuning of the training schedule, which, while effective in certain benchmarks, offers limited new scientific insight into the mechanics of representation learning.

- Sensitivity to Hyperparameters (Reviewer xHep): The effectiveness of HALD is highly dependent on the timing of the switches between stages. The authors' rebuttal suggests a balanced split (75/75 epochs), but this lacks robustness across datasets with different convergence properties. This sensitivity suggests that the method may require extensive per-dataset tuning, limiting its practical "plug-and-play" utility in diverse real-world scenarios. In addition, the problem addressed is specific to a niche setting: distillation without teacher access during training. If the teacher model is available (even a small one), the "drift" can be mitigated more effectively online.

**Reviewer Scores:**

Reviewer 6Y9U (Initial: 2): While the rebuttal addressed some empirical doubts, this reviewer’s fundamental concern about the "global label dominance" viewpoint and the "control-variate" theoretical mismatch remains a strong barrier to a high score.

Reviewer xHep (Initial: 4): This reviewer will likely appreciate the extra VGG/ViT experiments but will remain skeptical about the "strangeness" of the performance degradation in the baselines, viewing the method as a specific "recipe" rather than a generalizable contribution.

Reviewer x3At (Initial: 8): Likely to remain positive.

Core Reasons for Reject: The theoretical section serves more as a motivational backdrop than a formal proof for the proposed 3-stage pipeline. The transition from "gradient alignment" to a "soft-hard-soft switch" is not mathematically substantiated. And the method introduces new "timing" hyperparameters that increase the complexity of the training pipeline without a clear principled way to determine them across varying tasks.

---

### Decision · Program_Chairs · 2026-01-26

Reject